# Simultaneously enhancing the ultimate strength and ductility of high-entropy alloys via short-range ordering

Shuai Chen [1], Zachary H. Aitken[1], Subrahmanyam Pattamatta[2], Zhaoxuan Wu[2], Zhi Gen Yu [1], David J. Srolovitz [3✉], Peter K. Liaw [4✉] & Yong-Wei Zhang [1✉]

Simultaneously enhancing strength and ductility of metals and alloys has been a tremendous challenge. Here, we investigate a CoCuFeNiPd high-entropy alloy (HEA), using a combination of Monte Carlo method, molecular dynamic simulation, and density-functional theory calculation. Our results show that this HEA is energetically favorable to undergo short-range ordering (SRO), and the SRO leads to a pseudo-composite microstructure, which surprisingly enhances both the ultimate strength and ductility. The SRO-induced composite microstructure consists of three categories of clusters: face-center-cubic-preferred (FCCP) clusters, indifferent clusters, and body-center-cubic-preferred (BCCP) clusters, with the indifferent clusters playing the role of the matrix, the FCCP clusters serving as hard fillers to enhance the strength, while the BCCP clusters acting as soft fillers to increase the ductility. Our work highlights the importance of SRO in influencing the mechanical properties of HEAs and presents a fascinating route for designing HEAs to achieve superior mechanical properties.

[1] Institute of High Performance Computing, A*STAR, Singapore, Singapore. [2] Department of Materials Science and Engineering and Hong Kong Institute for Advanced Study, City University of Hong Kong, Hong Kong, SAR, China. [3] Department of Mechanical Engineering, The University of Hong Kong, Hong Kong, SAR, China. [4] Department of Materials Science and Engineering, The University of Tennessee, Knoxville, TN, USA. ✉email: srol@cityu.edu.hk; pliaw@utk.edu; zhangyw@ihpc.a-star.edu.sg

Conventional metallic alloys typically consist of one or two principal elements with minor additions of other elements (e.g., titanium alloys[1], magnesium alloys[2], and aluminum alloys[3]). In 2004, Yeh et al.[4] and Cantor et al.[5] proposed a class of alloys with five or more metallic elements of equal/near-equal atomic concentrations; these are now widely known as compositionally complex or high-entropy alloys (HEAs). Since then, HEAs have drawn increasing attention from both the scientific and industrial communities[6,7] for their mechanical properties, including an uncommon balance between strength and ductility[8] (important for high-performance structural material applications). Many experimental studies demonstrated that HEA strength and ductility are highly dependent upon their micro-/nano-structures[9,10]. Understanding the structure-property relations of HEAs enables rational HEA design[11].

Several approaches have been proposed to tailor HEA microstructures to promote the strength-ductility synergy[8]; e.g., modifications include introducing $Ll_2$ intermetallic nano-precipitates into face-center-cubic (FCC) (FeCoNi)$_{86}$-Al$_7$Ti$_7$[12] or Al$_{0.5}$Cr$_{0.9}$FeNi$_{2.5}$V$_{0.2}$[13], dual-phase FCC/hexagonal-close-packed (HCP) microstructures in Fe$_{50}$Mn$_{30}$Co$_{10}$Cr$_{10}$[14], or Cr$_{20}$Mn$_6$Fe$_{34}$Co$_{34}$Ni$_6$[15], nanoscale-disordered interfaces between adjacent micrometre-scale superlattice grains in Ni$_{43.9}$Co$_{22.4}$Fe$_{8.8}$Al$_{10.7}$Ti$_{11.7}$B$_{2.5}$[16], lamellar eutectic AlCoCrFeNi$_{2.1}$ with inter-granular B2 precipitates[17], body-centered tetragonal nano-precipitates into body-center-cubic (BCC) Ti$_{38}$V$_{15}$Nb$_{23}$Hf$_{24}$[18], heterogeneous non-recrystallized/recrystallized microstructures in Al$_{0.1}$CoCrFeNi[19], and forming ordered oxygen complexes in TiZrHfNb[20].

Recently, Ding et al.[21] synthesized FCC CoCrFeNiPd and CoCrFeNiMn HEAs. The former exhibited a higher yield strength than the latter with comparable tensile ductility. They proposed that the essential difference between these two HEAs was associated with atomic segregation and short-range ordering (SRO); both were enhanced in CoCrFeNiPd, as compared with CoCrFeNiMn. The idea was that SRO tends to create resistance to deformation/slip, leading to larger dislocation glide resistance in the more ordered CoCrFeNiPd than in CoCrFeNiMn HEAs. Theoretical work by Yin et al.[22] suggested that the strengthening was due mainly to the large atomic/misfit volume of Pd in CoCrFeNi. Ma et al.[23] identified a number of unusual features associated with dislocations/slips in HEAs that were related to lattice distortion and/or local chemical order. SRO also occurs in medium-entropy alloys (MEAs). The atomic structure of a CoCrNi MEA indicates that Cr favors Ni and Co neighbors, reducing the electrical and thermal conductivities[24] and increasing the stacking-fault energies and hardness[25].

Since segregation and SRO are all atomic-level phenomena, it is a challenge to directly interrogate SRO formation/evolution in experiments for demanding on-the-fly experimental decision-making, based on automated characterization[26]. Therefore, we turn to atomic-scale simulation tools to deepen our understanding of the SRO/mechanical behavior relationship in HEAs. The longer-time and larger-system size accessible with atomistic-simulation techniques, such as molecular dynamics (MD)[27,28] and Monte Carlo (MC)[29–31], can provide insight into nanoscale phenomena in HEAs that are not easily accessible via first-principle calculations[32,33]. For example, Li et al.[27] performed MD simulations of a nanocrystalline CoNiFeAl$_{0.3}$Cu$_{0.7}$ HEA with grain sizes comparable to those in the experiments of Fu et al.[34] and showed that the higher yield strength of the CoNiFeAl$_{0.3}$Cu$_{0.7}$ HEA (1.8 GPa compared with most other FCC HEAs, 0.2–0.6 GPa)[35–37] was associated with a strain-induced FCC to BCC phase transition.

Hybrid MC/MD simulations have proven to be a powerful tool to explore the effects of atomic segregation and SRO on the mechanical properties of MEAs and HEAs[29–31]. For example, Jian et al.[29] used hybrid MC and MD simulations to investigate the role of lattice distortion (LD) and SRO on the formation and evolution of dislocations in a CoCrNi MEA. These simulations suggested that yield strength was controlled by the strain required to nucleate Shockley partial dislocations; LD lowered this strain, while SRO increased it. The hybrid MC/MD simulations of Li et al.[30] demonstrated that increasing SRO in the CoCrNi MEA increased the ruggedness of the energy landscape, thereby raising barriers to dislocation activity and increasing strength. Short-range ordering in HEAs is more complex than that of MEAs (with the increased number of distinct atomic pairs in HEAs[31]), suggesting that HEAs provide a richer environment for tailoring their orders and microstructures to achieve superior strength and ductility. Is it possible to introduce SRO in HEAs to enhance both strength and ductility? If so, what microstructures are required? And, what is the underlying mechanistic reason for these increases? These critical questions are fundamental to rational HEA-processing strategies and alloy design for high-performance structural metals.

Here, we perform hybrid MC/MD simulations to explore the effect of SRO on the mechanical properties of an equiatomic CoCuFeNiPd HEA. Monte Carlo is employed to exchange atoms of different types to lower the alloy energy by optimizing SRO, and MD is employed to both relax the local atomic structure/displacements and to simulate tensile deformation. The Warren-Cowley parameters (WCPs) for all elemental pairs are used to describe SRO. Our MC/MD simulations show that the SRO development in the CoCuFeNiPd HEA is energetically favored, which is also validated by DFT calculations. This SRO is characterized by a wide range of local environments/atomic-scale configurations. Our MD tensile deformation simulations in this HEA with and without SRO indicate that the SRO enhances both the ultimate strength and ductility of the HEA. The underlying deformation mechanism is phase transformation prior to the ultimate tensile stress and dislocation slip posterior to the ultimate tensile stress. We explore the strain-induced phase transformations by investigating the local relative stability of FCC and BCC structures (sensitive to the local elemental concentration). The underlying mechanism responsible for the enhancement of both the ultimate strength and ductility in the HEA by SRO is described in terms of their mechanical responses of the three types of clusters: indifferent clusters play the role of a matrix, FCC-preferred clusters serve as hard fillers that enhance strength, and BCC-preferred clusters act as soft fillers that increase the HEA ductility.

## Results

**Atomic configurations and short-range ordering.** The variation of the potential energy per atom for the CoCuFeNiPd HEA with MC swap/MD relaxation (iteration) at 300 K is shown in Fig. 1a. These data show that the potential energy decreases with iterations, suggesting the formation of energetically more favorable atomic configurations. The potential energy per atom in the HEA at 0, $2 \times 10^6$, and $4 \times 10^6$ iterations at 300 K (0, 2, and 4 M samples) is $-4.090$, $-4.111$, and $-4.113$ eV/atom, respectively. The evolution of the corresponding atomic structure and SRO (i.e., the WCPs) is presented in Fig. 1b–d (atoms colored according to element type) and Fig. 1e–g (negative values of the WCP indicate favorable atomic pairs). Figure 1b, e show that the initial atomic configuration (the 0 samples) is nearly fully random, and all WCPs are near zero (no SRO).

Despite the fact that the change in the mean energy per atom is relatively small (0.023 eV/atom) following the SRO development, the effect on the alloy microstructure is large; cf. the microstructures in Fig. 1b–d. Clearly, the resulting SRO indicates clear

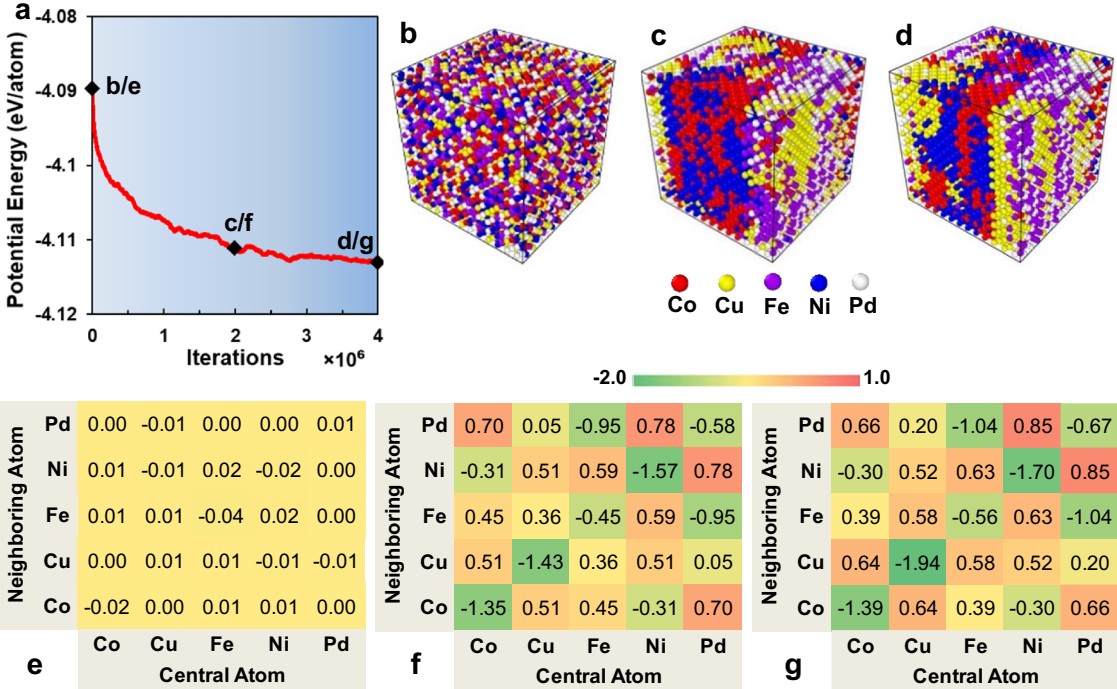

**Fig. 1 Potential energy, atomic configurations, and Warren-Cowley parameters of the CoCuFeNiPd HEA during iterations at 300 K. a** Variation of potential energy with iterations. Atomic configurations colored according to element types and tables of Warren-Cowley parameters at 0 (**b** and **e**), $2 \times 10^6$ (**c** and **f**), and $4 \times 10^6$ (**d** and **g**) iterations, as indicated in (**a**).

segregation of the different elements and that some elements show much stronger segregation than others. This tendency is captured by the WCPs in Fig. 1f, g, where the propensity of some elements to form clusters (negative SRO) and others to favor neighbors of other types is clear. There is a strong tendency to form Cu-Cu (WCP = −1.94 in Fig. 1g), Ni-Ni (WCP = −1.70 in Fig. 1g), and Co-Co pairs (WCP = −1.39 in Fig. 1g). The WCP data also show that Fe-Pd pairs are favored (WCP < 0), and Ni-Pd pairs are disfavored (WCP > 0). Interestingly, Xu et al.[37] and Santodonato et al.[38] also observed Cu segregation in $Al_x$CoCr-CuFeNi HEAs, which they attributed to the addition of Al and the precipitation of a γ' or $B_2$ phase.

At first glance, it is surprising to see that such a small reduction of potential energy per atom (0.023 eV/atom) during the SRO formation is able to induce such a marked change in the microstructures of the HEA. To confirm such a small energy reduction due to the SRO, we estimate the energies of the system at different stages of SRO based on the cohesive energies (energy per atom) of all the elemental pairs and the total number of such pairs. To do so, DFT calculations are performed to determine the cohesive energies of the binary alloys in an $L1_2$ $AB_3$ and $BA_3$ structures [$E_C(AB_3)$ and $E_C(BA_3)$] for all binary combinations of {Co, Cu, Fe, Ni, and Pd}, from which, the average cohesive energy for the binary AB alloy can be estimated based upon $E_{AB} = [E_C(AB_3) + E_C(BA_3)]/2$ and shown in Supplementary Table 1a, where A, B ∈ {Co, Cu, Fe, Ni, Pd}. The numbers of atomic pairs in the CoCuFeNiPd HEA (denoted as $N_{AB}$) are also calculated for the 0 (Supplementary Table 1b), 2 M (Supplementary Table 1c) and 4 M (Supplementary Table 1d) samples. The average cohesive energies (energy per atom) for these three samples can be calculated as $E = \sum(E_{AB} \times N_{AB})/\sum N_{AB}$. The cohesive-energy reductions of the 2 M and 4 M samples are −0.015 and −0.018 eV/atom, respectively, which are very close to the potential energy reductions per atom from the MC/MD simulation (−0.021 and −0.023 eV/atom). These small differences primarily arise from the variations between the MD and DFT calculations in the

cohesive energies of the binary alloys. Here we calculated the cohesive energies of (nearest neighbor) atomic pairs from a surrogate structure that shares a cubic crystal lattice with the same coordination number/geometry as our FCC HEA. The simple rule of the mixture is the lowest level approximation of the many-body interactions/correlations in the HEA. In the present situation, this low-level approximation is sufficiently accurate to estimate the cohesive energies of the whole HEAs.

**Effects of SRO on mechanical properties of HEAs.** Uniaxial tensile deformation simulations were performed on the 0, 2, and 4 M samples at a strain rate of $2 \times 10^8 \, s^{-1}$ under 300 K or 1200 K. The corresponding uniaxial stress-strain curves are shown in Fig. 2a. Since the strain rate employed is very large, compared with typical experiments, the resultant stress-strain curves should be interpreted as indicative of trends rather than quantitative. The solid lines in Fig. 2a clearly show that the ultimate stress and corresponding strain of the 2 M sample (2M-300K: 4.8 GPa and 8.4%) are substantially higher than those of the 0 samples (0–300 K: 3.4 GPa and 6.0%) during tension at 300 K. The 4 M sample, which was equilibrated in the longer MC/MD simulation (stronger SRO), exhibited even larger ultimate stress and strain (4 M-300 K: 4.9 GPa and 9.6%) during tension at 300 K. The dotted curves in Fig. 2a present the stress-strain curves for the same samples, but tested at 1200 K. The high-temperature test data exhibited the same trends as those tested at 300 K but are shifted to lower ultimate stress and strain. Figure 2a also shows that Young's modulus is, as expected, considerably lower at 1200 K than at 300 K.

We use the ultimate stress and ultimate strain as metrics (not quantitative values) for relative comparison of strength and ductility. Comparing the ultimate stress/strain results in Fig. 2a and the WCPs in Fig. 1e–g, it is clear that both strength and ductility are larger in the more ordered HEA sample. Atomic configurations, colored according to the structure of the local atomic environments[39], are shown in Fig. 2b, c for the 0 samples

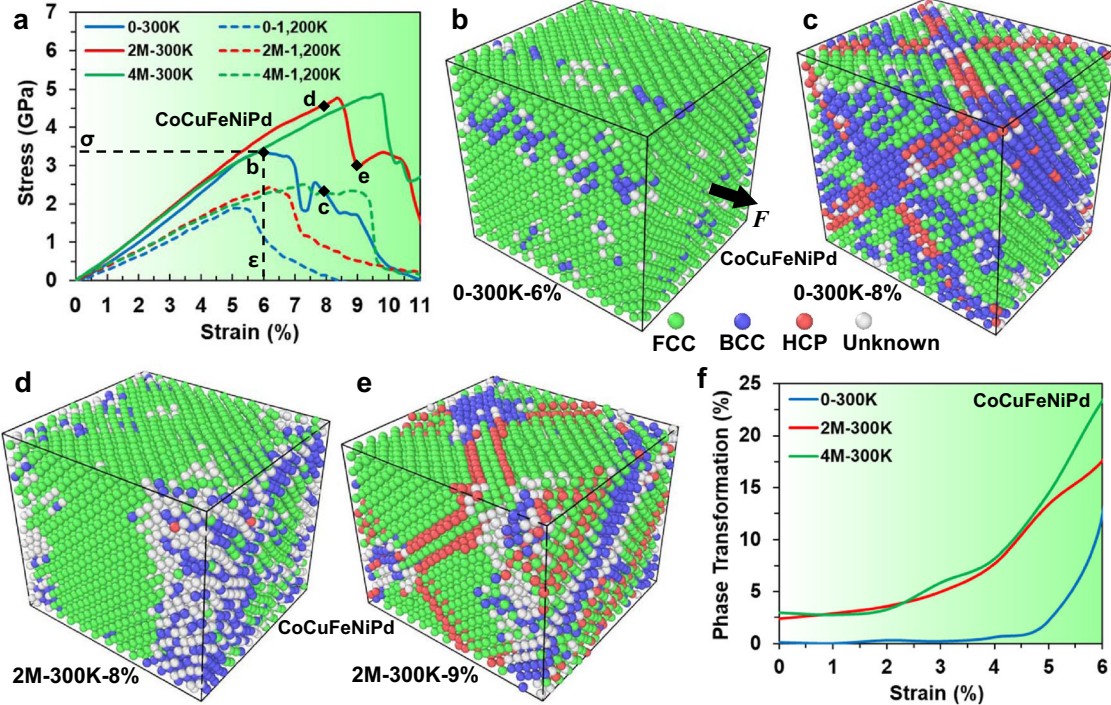

**Fig. 2 Stress-strain curves, atomic configurations, and phase transformations of the CoCuFeNiPd HEA with 0, 2 × 10⁶, and 4 × 10⁶ iterations at 300 K (0, 2, and 4 M samples) during tension at 300 K and 1200 K. a** Uniaxial tensile stress-strain curves for the 0, 2, and 4 M samples during tensions at 300 K (0–300 K, 2M-300K, and 4M-300K) and at 1200 K (0-1200 K, 2M-1,200 K, and 4M-1200 K). Atomic configurations of the 0 sample under 300 K tension at strains of (**b**) 6% (0-300K-6%) and (**c**) 8% (0-300K-8%) colored according to common neighbor analysis[39] (green, blue, and red indicate FCC, BCC, and HCP local environments). Atomic configurations of the 2 M sample under 300 K tension at strains of (**d**) 8% (2M-300K-8%) and (**e**) 9% (2M-300K-9%). **f** Variation of the BCC fraction (phase transformation) with strain for the 0, 2, and 4 M samples during tensions at 300 K.

at two different strains during a 300 K tensile test. Similar configurations for the 2 M sample are shown in Fig. 2d, e (see Supplementary Figure 1 for the 4 M sample data). Supplementary Figure 2a–f present atomic configurations near the ultimate strain to capture the structural transitions immediately prior to and posterior to the ultimate stress. The HCP structure regions seen in Fig. 2 and Supplementary Figures 1 and 2 are stacking faults associated with partial dislocation nucleation (Supplementary Figure 2b), which corresponds to a local (plastic) release of stress. The propagation of these partial dislocations occurs in relatively lower stress until blocked by phase boundaries or other dislocations (Supplementary Figure 2f), requiring an increase in stress for further motion, i.e., strain hardening. Figure 2 and Supplementary Figures 1 and 2 show that the dominant deformation mechanism is associated with FCC to BCC phase transitions in localized regions prior to the ultimate stress, while the post-ultimate stress deformation is dominated by dislocation slip within the untransformed FCC phase. Hence, the ultimate stress in the HEA corresponds to the nucleation and propagation of partial dislocations.

The observation that strain-induced phase transformations dominate the deformation in our CoCuFeNiPd HEA prior to the ultimate stress suggests that a relatively low tensile stress is required to nucleate BCC regions, compared with the larger stress required for partial dislocation nucleation. A previous experimental study showed that the high strength of a CoCu$_{0.7}$FeNiAl$_{0.3}$ HEA[34] arose primarily from grain-boundary strengthening and dislocation-strengthening mechanisms. In a subsequent MD study on the same HEA[27], it was found that when dislocation nucleation and emission were suppressed, strain-induced phase transformations from the FCC to BCC phase occurred. The variation of the BCC volume fraction (phase transformation) with

strain during tensile deformation is exhibited in Fig. 2f for the no-SRO and SRO samples under 300 K tension. These data show that no BCC-type structures were formed in the HEA without SRO for strains up to ~4%. Figure 2f also shows that a larger volume fraction of the BCC-type structure formed in samples with SRO (than no SRO) over the entire strain range, including at zero strain. The presence of small quantities of BCC-type structures in the samples with SRO developed in the MC/MD simulations (the 2 M sample: 2.4%, the 4 M sample: 3.0%) prior to deformation implies that this structure need not be strain nucleated but simply grows with increasing strain. These BCC-type structures form as a result of local lattice distortion-induced instabilities and are dispersed randomly throughout the whole sample. On the other hand, the sample with no SRO exhibits no initial BCC-like structure, and hence, nucleation is necessary prior to growth.

To confirm our observation that the observed ultimate strength and ductility are simultaneously enhanced via the SRO formation is not a strain-rate effect, we performed an additional set of simulations at a strain rate that is 10% that of the original simulation (i.e., $2 \times 10^7 \, s^{-1}$). The tensile stress-strain curves for the 0, 2, and 4 M samples during tensions at 300 K are plotted in Supplementary Figure 3a (along with the $2 \times 10^8 \, s^{-1}$ strain-rate results). The ultimate stress and associated ultimate strain for the 2 and 4 M samples are slightly smaller for the lower-strain-rate results, but still follow the same trends as reported in our higher-rate study. Similarly, examinations of the atomic configurations of the 2 M sample deformed at the lower strain rate for strains of 8 and 9% (Supplementary Figure 3b, c) indicate that the deformation mechanism remains unchanged upon the strain-rate reduction. For the low experimental strain rates ($10^{-4}-10^{-3} \, s^{-1}$), we employed Li et al.'s theory[40] for further exploration (Supplementary Discussion: Effect of strain rate on mechanical property of HEA).

**Phase transformations and cluster formation**. The results presented above demonstrate that the deformation mechanism prior to the ultimate stress for the HEA (with and without SRO) is associated with FCC to BCC phase transformations. To clarify the phase transformation processes observed above, we evaluate the relative stability of the FCC and BCC phases. In Step 1, we determine the cohesive energies of FCC and BCC ($E_{FCC}$ and $E_{BCC}$) random solid solutions as a function of the CoCuFeNiPd composition (in sufficiently large simulation cells to guarantee the convergence, $20 \times 20 \times 20$ nm$^3$ see Supplementary Figure 4). In Step 2, we determine the mean composition around each atom (including atoms in the first and second neighbor shells) in the HEA (with and without SRO). In Step 3, we calculate the local phase stability (i.e., $\Delta E_{FCC-BCC} = E_{FCC} - E_{BCC}$) based upon the local composition (Step 2) and the composition-dependent energies interpolated from the random solid-solution FCC and BCC phase data (from Step 1). All relevant data can be found in Supplementary Data 1.

Figure 3b, e show the atomic configurations colored according to their relative phase stability for structures without and with SRO. Negative $\Delta E_{FCC-BCC}$ indicates that for the local composition around an atom, an FCC structural environment is more stable than a BCC structure. Figure 3b demonstrates that in the system without SRO (the 0 samples), most of the local atom configurations prefer the FCC structure, while a few regions of a limited spatial extent prefer the BCC structure. On the other hand, in the presence of SRO (the 2 M sample), the regions that prefer BCC structures are of greater spatial extent and the degree to which BCC is more favorable than FCC is stronger (see Fig. 3e). Comparing Fig. 3d, e demonstrates that the stable FCC clusters ($\Delta E_{FCC-BCC} < 0$) tend to be located in Co-Ni-rich regions, while the less stable clusters ($\Delta E_{FCC-BCC} > 0$) are in regions that are enriched in Fe-Pd. Figure 3c, f show the atomic configurations colored according to the local structure following the tensile strain at 300 K. The locations where the FCC→BCC phase transformation appears are consistent with the clusters for which the composition suggests that BCC is more stable than FCC (see the black-circled-regions in Fig. 3e, f). Therefore, phase

transformation can be understood through the relative stability of the FCC and BCC structures, as determined by the local elemental concentrations. Regions of large composition variations from the mean occur preferentially in the systems with large SRO. In the system without SRO, few transformed regions are observed, and they are typically very small in their spatial extent, and occur more uniformly throughout the sample.

To gain a deeper understanding of the relative phase stability of the HEA, we calculated the $\Delta E_{FCC-BCC}$ distribution (in $10^{-2}$ eV intervals) of number fractions of atoms for the 0, 2, and 4 M samples (see Fig. 4a). These data clearly show that the $\Delta E_{FCC-BCC}$ distribution broadens during MC swaps and MD relaxations; i.e., the local preference for both BCC and FCC structures is enhanced by atom swapping. The energy reduction in forming stronger FCC-preferred configurations is offset by a similar tendency for BCC-preferred configurations. We can divide atoms into three groups according to their phase preference: FCC-preferred cluster (FCCP, $\Delta E_{FCC-BCC} \leq -3 \times 10^{-2}$ eV), BCC-preferred cluster (BCCP, $\Delta E_{FCC-BCC} \geq 1 \times 10^{-2}$ eV), and cluster indifferent to FCC/BCC phase (IND, $-3 \times 10^{-2}$ eV $< \Delta E_{FCC-BCC} < 1 \times 10^{-2}$ eV). The choice of $\Delta E_{FCC-BCC}$ thresholds is empirical.

The elemental concentrations for these three different phase-preference groups in the 2 M sample are tabulated in Fig. 4b; the FCCP atoms are primarily Co (29at%) and Ni (48at%), the BCCP atoms are Fe (31at%) and Pd (44at%), and IND atoms consist all species with a similar likelihood (Co: 16at%, Cu: 23at%, Fe: 27at %, and Pd: 27at%). The spatial distributions of FCCP, IND, and BCCP atoms are shown in Fig. 4c–e. Analysis of these structures shows that the IND clusters have weak SRO and form the matrix of the HEA. During the MC/MD iterations, some of the IND atoms transform to FCCP atoms with strong SRO, while some convert to BCCP atoms with strong SRO. We performed additional simulations with more iterations, starting from the 4 M sample to further enhance the SRO. The results are shown in Supplementary Figure 5 and Supplementary Discussion, indicating there is an optimal FCCP and BCCP fraction structure. The critical stress required for phase transformation in the sample with/without SRO is also analyzed and presented in

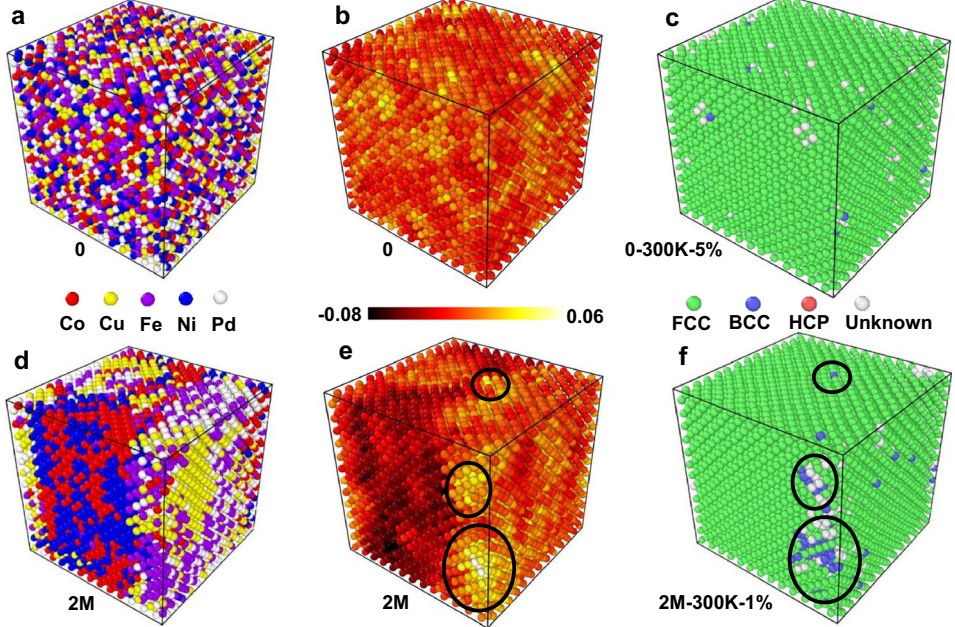

**Fig. 3 Atomic configurations, phase stabilities, and local structures of the CoCuFeNiPd HEA with 0 (0 samples) and 2 × 10$^6$ (2 M sample) iterations at 300 K.** Atomic configurations colored according to element type and phase stability ($\Delta E_{FCC-BCC}$) for the 0 samples (**a**, **b**) and the 2 M sample (**d**, **e**). **c** The 0 sample after a 5% strain (0-300K-5%), and **f** the 2 M sample after a 1% strain (2M-300K-1%) during tension at 300 K.

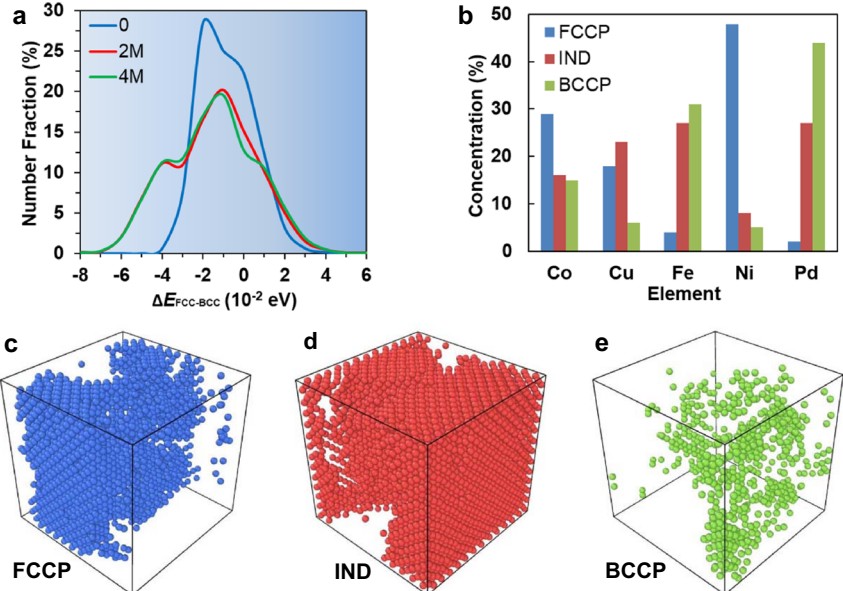

**Fig. 4 Number fractions, elemental concentrations, and spatial distributions of atoms with different phase preferences in the CoCuFeNiPd HEA. a** Distribution of atoms according to their phase stability $\Delta E_{FCC-BCC}$ for the 0, 2 M, and 4 M samples. **b** Elemental concentrations for the three different phase-preference groups in the 2 M sample. Spatial distributions of atoms by phase preference in the 2 M sample: (**c**) FCC-preferred (FCCP), (**d**) indifferent (IND), and (**e**) BCC-preferred (BCCP) clusters.

Supplementary Figure 6 and Supplementary Discussion (Optimal FCCP/BCCP fraction in HEAs).

**Deformation mechanisms responsible for the enhanced strength and ductility**. To characterize the deformation mechanisms responsible for the enhanced strength and ductility in terms of SRO in the HEA, we examine the stress and strain for these three phase-preference groups in the 2 M sample. The variation of average stresses (normal stress per atom parallel to the tensile axis) in these different phase-preference groups with strain during 300 K tension is shown in Fig. 5a. Atomic configurations colored according to their normal stress parallel to the tensile axis (0−10 GPa) at 2%, 4%, 6%, and 8% strains during loading are shown in Supplementary Figures 7a–c, and Fig. 5b, respectively, where dark colors (black/red) indicate low-stress levels, and bright colors (yellow/white) imply high-stress levels. These configurations qualitatively indicate that regions of high Co-Ni concentrations (i.e., FCCP clusters) tend to withstand high stresses. Figure 5a demonstrates that BCCP clusters and IND clusters resist similar stresses, while atoms in the FCCP clusters, on average withstand ~ 2 GPa/atom higher stresses than atoms in the BCCP and IND clusters when the strain increases from 2–10%. Clearly, FCCP clusters serve as hard fillers in the IND matrix that strengthen the HEA in this pseudo-composite microstructure.

The stress-strain curves of the 2 M sample during loading and unloading processes at 300 K are plotted in Fig. 5c. The distribution of plastic strains parallel to the tensile axis among the three phase-preference clusters in the 2 M sample after unloading to zero stress (marked in Fig. 5c) is shown in Fig. 5d (see Supplementary Figure 7e, f for the atomic configurations colored according to their phase structures and strain distributions). These data demonstrate that the BCCP clusters undergo a large plastic strain, while the FCCP clusters suffer relatively little plastic strain. Clearly, the BCCP clusters act as soft fillers in the IND matrix and enhance the ductility of the HEA within the pseudo-composite microstructure. It is interesting to note that some of the BCC domains undergo reversible transformation to

the FCC phase upon unloading. Since the simulations are isothermal, this phase transformation is strain-induced rather than thermoelastic. We performed additional simulations of bicrystal samples with grain boundaries, as shown in Supplementary Figures 8 and 9. The bicrystal sample with SRO also exhibits enhanced ultimate strength (Supplementary Discussion: Effect of grain boundary on mechanical property of HEA). We also explored the effect of annealing temperature on SRO and mechanical properties of HEAs (Supplementary Figures 10 and 11 and Supplementary Discussion: Effect of annealing temperature on SRO of HEAs).

## Discussion

**Comparisons with experiments**. Many strategies have been proposed to tailor structural heterogeneities to promote the strength-ductility synergy in HEAs[8]. Generally, these strategies are associated with the manipulation of grain-size heterogeneity, precipitation, dual-phase microstructures, interstitial complex, and short-range ordering (Table 1). One grain-size strategy was to combine non-recrystallized (1 to 20 μm grain size) and recrystallized grains (0.2–5 μm grain size) within a microstructure to mediate the strength-ductility trade-off in an $Al_{0.1}CoCrFeNi$ HEA[19]. A disordered FCC matrix with ordered $Ll_2$ nanoscale precipitates was employed in an $Al_{0.5}Cr_{0.9}FeNi_{2.5}V_{0.2}$ HEA[13] to increase strength while retaining ductility. A soft FCC lamella matrix with a hard intergranular $B_2$ (an ordered BCC) phase was designed in an $AlCoCrFeNi_{2.1}$ HEA[17] for the simultaneous strength-ductility enhancement. Tuning the phase stability by changing the Mn content led to a two-phase microstructure in $Fe_{50}Mn_{30}Co_{10}Cr_{10}$[14] and $Cr_{20}Mn_6Fe_{34}Co_{34}Ni_6$ HEAs[15]. Interstitial complexes were formed in TiZrHfNb HEAs by the introduction of oxygen or nitrogen. Oxygen enhanced both the strength and ductility while nitrogen increased the strength but decreased the ductility[20]. While the first four of these strategies focus on structural heterogeneities in HEAs over relatively large length scales (nanometers to many micrometers), the SRO-based approach focuses on the atomic scale. It provides a pathway for the manipulation of HEAs at the atomic level that leads to

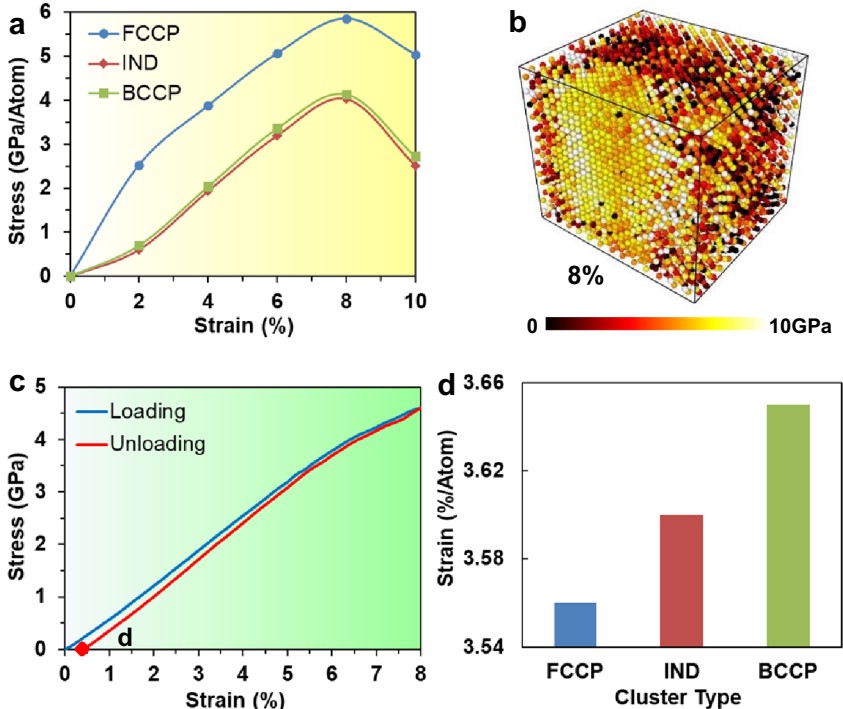

**Fig. 5 Average stress, stress distribution, stress-strain curve, and average strain of the 2 M sample during loading and unloading processes at 300 K.** **a** Variations of average normal stresses parallel to the tensile axis for atoms of different phase preference with strain in the 2 M sample during loading at 300 K. **b** Atomic configuration of the 2 M sample at a strain of 8% colored according to their stress distributions under 300 K loading. **c** Stress-strain curves of the 2 M sample during loading and unloading processes at 300 K. **d** Average atomic-level plastic strain parallel to the tensile axis for atoms of different phase preference in the 2 M sample after unloading to zero stress (marked in (**c**)).

**Table 1 Comparison of different approaches for the strength and ductility synergy in HEAs.**

| Heterogeneities | HEA | Microstructure | Strength | Ductility |
|---|---|---|---|---|
| Grain size | $Al_{0.1}CoCrFeNi$[19] | Non-recrystallized + Recrystallized grains | √ | √ |
| Precipitate | $Al_{0.5}Cr_{0.9}FeNi_{2.5}V_{0.2}$[13] | FCC + $LI_2$ | √ | × |
| | $AlCoCrFeNi_{2.1}$[17] | FCC + $B_2$ | √ | √ |
| Dual phase | $Fe_{50}Mn_{30}Co_{10}Cr_{10}$[14] | FCC + HCP | √ | √ |
| | $Cr_{20}Mn_6Fe_{34}Co_{34}Ni_6$[15] | FCC + HCP | √ | √ |
| Interstitial complex | TiZrHfNb[20] | $(TiZrHfNb)_{98}O_2$ | √ | √ |
| | TiZrHfNb[20] | $(TiZrHfNb)_{98}N_2$ | √ | × |
| Short-range ordering | CoCrFeNiPd[21] | Atomic segregation | √ | × |
| | CoCuFeNiPd[#] | Atomic segregation | √ | √ |

√ / × indicates increasing/decreasing (or no change) in a property. # identifies the HEA studied here.

strength-ductility synergy. This approach can, of course, be combined with larger-scale approaches (e.g., grain size, precipitates, etc.) to further optimize HEA mechanical response. In our HEA, SRO clusters have similar elemental concentrations and the same crystal structures as the HEA matrix. For example, FCCP, IND, and BCCP are SRO clusters that have similar elemental concentrations and the same FCC crystal structure. On the other hand, the crystal structure and chemical composition of a nanoscale precipitate in a HEA are well defined and generally differ from those of the HEA matrix. Hence, we do not consider the SRO clusters as precipitates.

The connection between SRO and the mechanical response of CoCrFeNiPd HEAs was previously noted; SRO was shown to increase the resistance to dislocation glide with a concomitant increase in the HEA strength (without compromising ductility)[21]. In the present CoCuFeNiPd HEA study, we demonstrated that SRO can lead to the formation of a pseudo-composite microstructure of BCCP, IND, and FCCP structures. Prior to reaching the ultimate stress, localized structural (FCC to BCC) phase transformations occurred that increase both the ultimate strength and ductility. This feature only occurred (at a significant level) because of the compositional SRO that developed upon annealing; this SRO development requires the atomic transport on a ~5 nm scale. In the earlier CoCrFeNiPd HEA study, the atomic transport on the requisite length scale did not occur, leading to measured compositional correlation lengths of only 1–3 nm[21]. The formation of SRO on annealing in our samples led to the formation of localized BCCP structures, which serve as FCC to BCC phase transformation nuclei on the application of stress. These phase transformations provide the local heterogeneities necessary for dislocation formation in our single-crystal system. While grain boundaries in HEAs may serve as dislocation nucleation sites in polycrystals, the density of such sites is small, compared with that of the high density of heterogeneities induced by the highly localized BCC phases that form on annealing. We note that phase transformation has been observed in an earlier

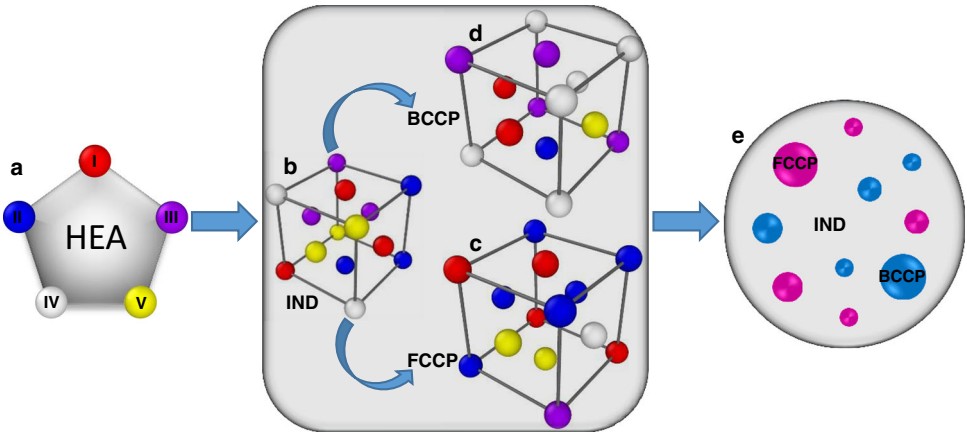

**Fig. 6 Schematic diagram illustrating the strategy to design HEAs with simultaneously enhanced strength and ductility via short-range ordering. a** Design a HEA with five elements (I, II, III, IV, and V). Formations of (**b**) IND, (**c**) FCCP, and (**d**) BCCP clusters via short-range ordering. **e** The IND clusters serve as a matrix with the FCCP clusters serve as hard fillers to enhance the strength and BCCP clusters act as (plastically) soft fillers to increase the ductility.

simulation study[27]. This trend supports the deformation mechanism observed here.

**Strategy to design HEAs with large strength and ductility via SRO**. The present work indicates that the development of SRO (with atom swapping on a scale of ~5 nm) in HEAs can induce structural heterogeneities on a sufficiently small scale to achieve excellent mechanical properties. Figure 6 provides a schematic illustration of the proposed HEA-design strategy for the strength and ductility enhancement via SRO. The central point is that SRO can produce a pseudo-composite microstructure of FCCP, BCCP, and IND domains. In this composite microstructure, the IND domains play the role of a matrix, FCCP domains serve as hard fillers that enhance strength, and BCCP domains act as soft fillers that enhance ductility. Using the five elements, labeled I, II, III, IV, and V as shown in Fig. 6a as an example, we illustrate the required conditions to form such a composite microstructure. First, all five elements tend to form a stable FCC structure in the random solid-solution phase (i.e., IND domain as shown in Fig. 6b); second, highly stable FCCP domains are able to form via SRO (Fig. 6c); and finally, BCCP domains can also form (Fig. 6d) while in the BCCP domains, the FCC is metastable (i.e., BCC is the thermodynamically-stable phase). Such a HEA tends to form with a matrix, which is nearly indifferent to FCC/BCC phase (this is the IND domain) with the stable FCCP domains acting as hard fillers (for strength) and BCCP domains acting as soft fillers (for ductility), thus leading to a pseudo-composite with both very high strength and ductility (Fig. 6e).

While the Gibbs phase rule indicates that more phases can co-exist in equilibrium in an alloy with a larger number of components, multiple phases can co-exist in binary and ternary alloys, but such microstructures are fundamentally unstable. We have shown that the SRO-induced composite microstructure consists of three types of clusters: FCCP, IND, and BCCP clusters. The formation of such a composite microstructure is associated with the SRO (previously identified[41]); the SRO is associated with a large chemical-affinity disparity and high chemical-element exclusivity amongst constituent chemical elements. Clearly, multiple elements are required to form these different atomic configurations and structures; e.g., the initial FCC phase in a random solid solution and the subsequent FCCP, BCCP, and IND clusters that form as a result of SRO. Intrinsic FCC stabilizers are also essential to form the BCC phase during the strain-induced

phase transformation from BCCP structures. These different requirements on the formation of such a composite micro-structure require multiple elements. Hence, it is expected that the formation of such a composite microstructure is much more prevalent in HEAs than in simple binary or ternary alloys. Examination of the two samples with the same elemental concentrations as the FCCP and BCCP clusters (Supplementary Figure 12a) verifies that FCCP is the hardening domain. Random FCC NiCo alloys with pre-existing dislocation have yield strengths dominated by solute-misfit[42]. However, NiCo has a near-zero misfit such that random NiCo alloys exhibit a very low yield strength[42]. In our simulations, the five-element FCCP clusters (29 Co, 18 Cu, 4 Fe, 48 Ni, 1 Pd (at%)) exhibit SRO not present in random NiCo alloys. Besides, FCCP clusters do not have pre-existing dislocation. Hence, it is no surprise that the FCCP clusters in the HEAs exhibit quite different mechanical properties from random NiCo alloys.

The elastic interaction between FCC and BCC phases plays an important role in the work-hardenability of HEA via the deformation-induced phase transformation. Supplementary Figure 7d shows the atomic configurations of two domains in the 2 M sample at the strain of 8% colored according to phase structures or stress distributions. Prior to dislocation nucleation, we observe that the elastic interaction between the FCC and BCC phases causes a stress concentration at the FCC and BCC phase interface (Supplementary Figure 7d), which in turn drives the FCC to BCC phase transformation. With increasing external strain, the elastic interactions lead to a larger stress concentration at the interface, inducing more phase transformation, and thus contributing to the work-hardenability (i.e., the stress increases with the phase transformation growing in stress-strain curves of Fig. 2a). At a critical moment, the nucleation of dislocations occurs at the boundary between the FCC and BCC phases (Supplementary Figure 2), indicating that the elastic interactions are also the driving force for dislocation nucleation. The evolution of a BCC domain is shown in Supplementary Figure 13, which indicates that the BCC domain expands and phase boundaries move towards the FCC phase prior to dislocation nucleation (Supplementary Figures 13b–f) due to the increasing phase transformation. After dislocations are nucleated (due to the elastic interactions of FCC and BCC), the BCC domain shrinks and phase boundaries move towards the BCC phase (Supplementary Figures 13f, g) due to the stress reduction and dislocation nucleation/propagation.

We performed iterative MC and MD simulations to examine the evolution of SRO in the CoCuFeNiPd HEA and MD simulations to investigate the effect of SRO on mechanical behavior. During the formation of SRO, some small regions transform to BCC-preferred structures (BCC is more stable than FCC), while others become very strongly FCC-preferred structures (FCC is much more stable than BCC). The resulting structure is a pseudo-composite microstructure consisting of very stable FCC-preferred (FCCP) regions and BCC-preferred (BCCP) regions in a matrix of marginal stability. Tensile-deformation simulations indicate that the ultimate strength and ductility of the HEA are both enhanced by the development of the SRO. The origin of these enhancements can be traced to the SRO-induced pseudo-composite microstructure that forms; this consists of both FCCP hardening domains and BCCP toughening domains in a matrix of marginal phase stability. We propose a strategy for simultaneously achieving excellent strength and ductility via SRO in HEAs. The present study not only reveals the role of SRO in governing the strength and ductility of HEAs, but also provides guidelines for rationally designing HEAs with high-performance mechanical properties for engineering applications.

## Methods

**Atomic potentials**. Hybrid MD and MC simulations are performed, using the large-scale atomic/molecular massively parallel simulator (LAMMPS) package[43] with Zhou et al.'s[44,45] embedded atom model (EAM) potential parameters. These potential parameters have been employed in earlier HEA simulations with reasonable results[27,46]. We have checked the reliability of the atomic potential used in the study from three different aspects: (1) lattice constants, (2) cohesive energies, and (3) melting points. A good agreement in the lattice constants, cohesive energies (Supplementary Table 2) and melting points (Supplementary Figure 14) obtained using the atomic potential and from other sources is observed, thus validating the reliability of the atomic potential used in the study (detailed analysis and comparison are presented in Supplementary Discussion: validating the reliability of the atomic potential).

**Atomic swaps and relaxations**. The lattice constant of the FCC CoCuFeNiPd was initially set at 3.6 Å in a simulation cell containing 8788 atoms, which is oriented with [100], [010], and [001] directions aligned respectively with x, y, and z axes. The initial sample was constructed by populating atomic sites randomly with Co, Cu, Fe, Ni, and Pd subject to the near-equiatomic elemental composition constraint, i.e., 1,757 atoms each for Co, Cu, Fe, and Ni and 1760 atoms for Pd. The HEA elemental distribution was optimized by performing MC site-occupancy swaps between pairs of sites at a temperature of 300 and 1200 K. The acceptance of each MC swap conforms to the Metropolis criterion[47], i.e., if the system energy following the swap attempt, $i + 1$, $E(i + 1)$, is lower than that following the previous successful swap, $E(i)$, the MC swap is accepted. Otherwise, it is accepted with probability

$$P = e^{\frac{E(i+1)-E(i)}{kT}}, \tag{1}$$

where $k$ is the Boltzmann constant, and $T$ is the temperature. If a uniformly generated random number in the range of (0,1), $R$, is lower than / equal to $P$, the MC swap is accepted. Otherwise, it is rejected. The MC steps are interchanged with MD relaxations to efficiently converge site occupancy and atomic displacements (we perform 100 MC swaps followed by up to 100 MD relaxations per iteration).

Energy calculations for phase stability. To characterize the relative phase stability, the cohesive energies of FCC and BCC structures with random site occupancies were calculated in a $20 \times 20 \times 20$ nm$^3$ simulation cell (periodic boundary conditions in three ⟨100⟩ directions) for a large set of compositions, where the atoms were relaxed using a conjugate gradient minimization of the potential energy (zero stress in all three ⟨100⟩ directions). The associated energies are summarized in Supplementary Data 1. This database was interrogated (and interpolated) to provide FCC and BCC energy differences at compositions around (within the first and second neighbor shells) individual atoms from the MC/MD iterations.

**Characterization of short-range ordering**. The Warren–Cowley parameter (WCP) was employed to characterize the SRO in the 1st-nearest neighbor shell[48]:

$$WCP_{mn} = 1 - Z_{mn}/(\chi_n Z_m) \tag{2}$$

where $Z_{mn}$ is the number of n-type atoms around m-type atoms, $Z_m$ is the total number of atoms around m-type atoms, and $\chi_n$ is the atomic fraction of n-type atoms in the HEA. If $WCP_{mn} = 0$, mn-type pairs are randomly distributed (no SRO). If $WCP_{mn} < 0$, mn pairs are more abundant than random, while if $WCP_{mn} >$

0, mn-type pairs occur less often than random. To eliminate the influence of lattice distortions in the calculation of the WCPs, the atomic site occupancies are mapped to the corresponding perfect FCC structure (no lattice distortion; all atoms on regular lattice sites). The WCP calculations were performed on this lattice by counting the elemental types of 1st-nearest neighbors.

**Tensile deformation**. Prior to applying tensile deformation, the CoCuFeNiPd HEA was thermally equilibrated at 300 K or 1,200 K for 0.2 ns, via MD simulations in an isothermal-isobaric (constant number of particles, $N$, constant pressure, $P$, and constant temperature, $T$, i.e., NPT) ensemble (zero stress in all three ⟨100⟩ directions). Uniaxial tensile deformation was applied in the x-direction at a strain rate of $2 \times 10^8$ s$^{-1}$ or $2 \times 10^7$ s$^{-1}$ for 1.0 ns or 10.0 ns at each equilibration temperature. Strain rates of $2 \times 10^8$ and $2 \times 10^7$ s$^{-1}$ have been achieved, e.g., under laser shocking[49,50]. During tensile deformation, the NPT ensemble was employed in the y- and z-directions to maintain zero lateral pressure (i.e., constant uniaxial strain rate). Periodic boundary conditions were applied in all three directions. The integration time step was 1 fs, and the total simulation time (including thermal equilibration) was 1.2 ns (at a strain rate of $2 \times 10^8$ s$^{-1}$) or 10.2 ns (at a strain rate of $2 \times 10^7$ s$^{-1}$). OVITO[39] was used to visualize atomic configurations and analyze simulation results by identifying phase structures (common neighbor analysis) and calculating atomic strains.

**Density-functional-theory (DFT) calculations**. Density-functional-theory calculations were performed to determine the cohesive energies of all A-B (and A-A) elemental pairs in binary $L1_2$ AB$_3$ alloys for all A-B (and A-A) binary combinations {Co, Cu, Fe, Ni, and Pd}, using the Vienna ab initio simulation package (VASP)[51] with a plane-wave basis and projector augmented wave (PAW) potentials[52,53]. The Perdew, Burke, and Ernzerhof (PBE) exchange-correlation energy functional within the generalized gradient approximation (GGA) was employed[54]. For the binary-alloy, $L1_2$ AB$_3$, reference system, the cohesive energy is:

$$E_c(AB_3) = \frac{1}{4}\left[E_g(A) + 3E_g(B) - E_b(AB_3)\right] \tag{3}$$

where $E_b(AB_3)$ is the energy of the fully-relaxed AB$_3$ structure, and $E_g(A)$ and $E_g(B)$ are the energies of isolated A and B atoms in their ground states, respectively.

To compute the ground-state energies of the symmetry-broken spin-polarized magnetic ground state of an isolated atom, DFT calculations were performed for a single atom in a large cubic cell ($14 \times 14 \times 14$ Å$^3$) with periodic boundary conditions (PBCs) to avoid interactions with its periodic images. Relaxation runs were done by performing spin-polarized calculations and by allowing both the ionic positions, cell volume, and cell shape to relax. The convergence tolerance for the electronic self-consistency was set at $10^{-8}$ eV, and the plane-wave energy cutoff was 800 eV. The bulk energies of the AB$_3$ $L1_2$ alloy were calculated by fully relaxing the ion positions and periodic-cell lattice parameters to electronic self-consistency with a tolerance of $10^{-8}$ eV and a force convergence tolerance of $10^{-4}$ eV/Å. A plane-wave energy cutoff of 520 eV and k point mesh of $15 \times 15 \times 15$ per cell were applied for all calculations.

## Data availability
The cohesive-energy data of FCC and BCC structures generated in this study are provided in Supplementary Data 1.

## Code availability
The information of open-source software and input files for simulations that support the findings of this study are available in the Supplementary Information.

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

## Acknowledgements
S. C., Z. H. A., Z. G. Y. and Y.-W. Z. gratefully acknowledge the financial support from the Agency for Science, Technology and Research (A*STAR) under grant AMDM A1898b0043, and the use of computing resources at the A*STAR Computational Resource Centre and National Supercomputer Centre, Singapore. P. K. L. gratefully acknowledges the support of (1) the National Science Foundation (DMR-1611180 and 1809640) with program directors, Drs. J. Yang, G. Shiflet, and D. Farkas and (2) the US Army Research Office (W911NF-13–1-0438 and W911NF-19–2-0049) with program managers, Drs. M. P. Bakas, S. N. Mathaudhu, and D. M. Stepp. The contributions of D. J. S. and S. P. in this paper were fully supported by a grant from the Research Grants Council of the Hong Kong Special Administrative Region, China (Project No. 11211019).

## Author contributions
S. C. and Y.-W. Z. designed the research. S. C., Z. H. A. and Y.-W. Z. performed the molecular dynamics and Monte Carlo simulations. S. P., Z. W., Z. G. Y. and D. J. S. performed DFT calculations. D. J. S., P. K. L. and Y. W. Z. provided the supervision of all studies. S. C., Y.-W. Z., D. J. S. and P. K. L. wrote the paper. All authors contributed to the discussion and revision of the paper.

## Competing interests
The authors declare no competing interests.
