## [Peer Review File · Nature Communications]

Title: Simultaneously Enhancing the Ultimate Strength and Ductility of High-Entropy Alloys via Short-Range OrderingREVIEWER COMMENTS

Reviewer #1 (Remarks to the Author):

This paper shows that, in the CoCrFeNiPd HEA with FCC structure, there can be a composite-like structure composed of parts having different phase stability due to the formation of SRO, and some of the parts can transform into BCC phase during deformation with high strain rate, resulting in superior strength-ductility balance. Although the contents of the manuscript were interesting for me, there are several issues the authors need to address as follows.

(1) Consistency with experimental reports

This may be a typical criticism to many atomistic simulations. As far as I know that there is no experimental report on the CoCrFeNiPd HEA to date. Therefore, it is difficult to guarantee that the phenomena the authors reported can actually happen in reality. One suggestion is to cast an actual ingot of the HEA and make a very simple heat treatment to see whether the elemental partitioning like the authors reported actually happens or not. In the manuscript, the authors cited Z. Fu et al. (Ref. [32]) and other literature several times to support their results. However, the experimental conditions in the literature, such as chemical composition, processing conditions, and strain rate, are totally different. In addition, some of them seemed to be cited with inappropriate context. For instance, the authors cited Z. Fu et al. (Ref. [32]) to say that the nucleation of BCC phase could be easier than nucleating partial dislocations, but I don't think Z. Fu et al. argued such a thing in their paper. In my opinion, the authors should review the references they cited again to confirm the validity of their way to cite other literature.

(2) Universality of the phenomena

I think one of the most important parts of the manuscript is the formation of a composite-like structure due to SRO. Although the author focused on the HEA in the manuscript, I guess similar phenomena can occur not only in HEAs but simple binary or ternary systems. Thus, I wonder whether the authors found something special in the HEA or they wanted to say those are very common phenomena that we can observe in various alloy systems. Also, the authors employed a simple rule of mixture using DFT results to predict the local phase stability of the HEA, and the results appeared to be consistent with the MD results. Why can we use such a very simple rule of mixture to predict atomistic properties? Is there any limitation of this methodology? The authors should address these points in the manuscript.

(3) Role of composite-like structure

Many people say deformation-induced phase transformation can increase the work-hardenability of metallic materials, but, in my opinion, things may not be simple. I would like the authors to ask why the work-hardenability of the HEA increased in detail. One thing may be elastic interaction between FCC and BCC phase (or FCC phase parts having different chemical composition). I speculated that the parts having different elastic properties were dynamically formed and increased internal stress in the HEA due to the elastic interaction. Another thing can be the contribution of dislocations. It seemed to me there was some amount of plastic deformation in both FCC and BCC phases. I wonder how defects multiplied

and how they were different depending on the crystal structure or stability of the parts. In addition, these two factors should have interactions, and there can be other contributions like the effect of transformation strain. The authors should comment on these aspects.

Reviewer #2 (Remarks to the Author):

This work demonstrates a new effect of the SRO in HEAs, i.e., SRO changes the local concentration, which makes the bcc/fcc-preferred region more stable than in the ideal random state. This helps the formation of pseudo-composite microstructures (FCCP, IND, BCCP) when loaded, leading to better strength and ductility simultaneously. This is an interesting work and contributes a new perspective to the recent hot topic, "SRO in HEAs". I recommend publishing it when the following questions are resolved in the coming revision.

1.

Figure 2f shows that even at zero loadings, there are bcc regions in the 2M and 4M samples. Then how did you compute the WC-SRO parameter? How to define the neighboring shells?
In Figure 1e-g, why the tables are not exactly symmetric?

2.

One of the key concepts in the draft is that SRO produces a pseudo-composite microstructure, leading to better mechanical properties as compared to the random state. But I think we need to be more careful about this statement.

As shown in Figure 2b-c, even the ideal random state (sample 0) can have the composite microstructure at a strain of 6%, with the BCC fraction of ~12%. Therefore, it seems that the effect of SRO is to (1) make the BCCP domain easier to happen, (2) increase the fraction of the BCC phase.

Then my question is why more BCCP is better for mechanical properties, as compared to the random state? Is there any optimal fraction of BCCP?

It seems insufficient to only discuss the composite-based mechanism itself, as it applies in both random and SRO alloys. It would be better to add more analysis and discussions about the effect of SRO on the alloy mechanical properties, which is the reason why we shall use SRO to improve the alloys, instead of using just a random alloy.

3.

The fcc alloys usually have lower strength and more ductility as compared to the bcc alloys. See reference:

<https://doi.org/10.1016/j.actamat.2020.01.062>

However, it is quite the opposite in the FCCP and BCCP phases, as shown in Figure 5a. The FCCP phase is the hardening domain and Ni-Co-rich, but the NiCo alloy has a very low yield strength as shown in the reference above. Is there any discussion for that?

4.

The MC simulation was conducted at 300 K. In practice, such a low annealing temperature would prohibit the kinetic of diffusion. It would be better to add some discussion about the feasibility of the composite-based mechanism proposed in this work.

Best wishes,
Binglun Yin

Reviewer #3 (Remarks to the Author):

In this paper the authors report on a study of the atomic structures and mechanical behavior of a CoCuFeNiPd high entropy alloy (HEA). Unlike other well studied approaches that have been used to promote strength-ductility synergy in HEAs, such as precipitation, recrystallization, tuning the chemistry, and/or adding elements such as oxygen nitrogen and others authors of this study propose a strategy that involves short range order (SRO). In principle, SRO can increase the resistance to dislocation glide thereby yielding an increase in strength. This can be particularly effective when the microstructure contains a mixture of phases. Interestingly, the authors use an approach that involves a combination of the Monte Carlo method together with molecular dynamics and density functional theory. They use atomic simulations as a tool to understand both segregation and short range order(SRO) in this HEA. The paper is very well written and provides insight into phenomena that would be very difficult to assess experimentally. Like with all simulations one always wonders what would happen if for example deformation would occur at realistic strain rates. There are a few points that should be addressed prior to publication:

1. In the Introduction, it is surprising that the current authors did not mention the two important references related to SRO in the complex alloy field. "Zhang, F. X. et al. Local structure and short-range order in a NiCoCr solid solution alloy. Phys. Rev. Lett. 118, 205501 (2017)." "Zhang, R.P. et al. Short-range order and its impact on the CrCoNi medium-entropy alloy. Nature volume 581, pages283–287(2020)."

2. In Page 7 (line 133 and 134), for Cu segregation in Al_xCoCrCuFeNi HEAs, the addition of Al contributes

to the precipitation of γ' in Reference 35 or B2 in Reference 36, which causes the Cu segregation in the matrix. I think the conclusion in this part is not convincing.

3. In Page 9 (line 160), the strain rate for the tensile simulation is as high as $2 \times 10^8/s$, it seems this work belongs to ultra-high strain rate region. Therefore, I am wondering if the current simulated results can represent the mechanical properties of alloys under normal tensile test (10^{-4} - $10^{-3}/s$).

4. Also about this ultra-high strain rate, the recent work by Zhao et al. (Sci. Adv. 2021; 7: eabb3108) shows that even under strain rate of $6 \times 10^5/s$, an amorphous phase would appear. I am wondering what phase would be formed when the strain rate is 100 higher. Is it possible to form amorphous phase in this sample instead of BCC or HCP at such high strain rates?

5. In Fig. 2a, in the ss curves, it seems that the elastic strain is over 6%, which is similar to superelastic behavior. Since the theoretical maximum superelastic strain is closely related to the lattice parameter of austenite and martensite phases, what is the lattice parameters of the FCC matrix, strain or stress-induced BCC and HCP phases.

6. In Fig. 2f, for 2M and 4M sample, it seems that 5% of BCC phase already exists before tensile tests. Why?

7. In Fig. 2, it seems that the strength decreases with increasing amount of HCP phase. Why?

8. In Page 11 (line 196 and 197), the current authors claimed that strain-induced phase transformations dominate the deformation of the HEA, I would like to know if the current authors did any calculation for the critical stress for phase transformation.

9. In Fig.3, for the SRO and phase transformation in the current HEA, it seems that the Ni segregation helps to tune the martensite start temperature, which results in the heterogeneity of FCC-BCC transformation in the current samples. So the key to adjusting SRO and related phase transformation in the current HEA is the local concentration of Ni?

10. In Fig. 4, for the SRO with clusters of FCCP, IND and BCCP, it seems this kind of SRO is very similar to one with nano-scale precipitates, even in the element concentration. How can one clearly define SRO clusters and precipitates (3-5 nm) in the current HEAs?

11. In Fig. 5c and 5d, for the loading-unloading ss curves, this HEA undergoes reversible transformation (strain up to 8%). I am curious about the type of BCC phase? Is it thermoelastic?

12. In Table 1, for the first four types of HEAs, they are all polycrystalline and grain boundaries play an important role in tailoring their mechanical properties. However, the current HEA is single crystal, and grain boundaries are not taken into account. Based on their simulations and calculations, it seems that more severe element segregation would take place along grain boundaries. I would like to know if the SRO would still improve mechanical properties of this HEA when the grain boundaries are considered.

One of the questions that emerges is perhaps can the authors speculate what would be the morphology of the tensile deformation curves if the strain rate were to be reduced to a more normal level instead of the very high strain rate assumed by the simulations which corresponds to $2 \times 10^8 s^{-1}$.

The authors argue that in order to confirm the result that ultimate tensile strength and ductility are simultaneously enhanced via SRO formation is not related to strain rate and hence, they conducted simulations at strain rates that are 10% of the original simulation to support this argument. The question

arises as to why was the 10% of the original strain rate used as the baseline to validate the results?

Overall this is a well written paper that provides interesting and novel results of an approach that can be used to design HEAs in the future. Their results suggest that the development of SRO can induce structural variations on a sufficiently small scale to achieve outstanding mechanical properties. The authors are all well-known in their respective fields and it is my opinion that the paper as written, with some clarifications, represents an outstanding contribution to the literature.

In view of the above recommend acceptance once the above comments are addressed.

Response to Reviewer #1

Overall Comments: *This paper shows that, in the CoCrFeNiPd HEA with FCC structure, there can be a composite-like structure composed of parts having different phase stability due to the formation of SRO, and some of the parts can transform into BCC phase during deformation with high strain rate, resulting in superior strength-ductility balance. Although the contents of the manuscript were interesting for me, there are several issues the authors need to address as follows.*

Author reply: We thank the reviewer for her/his careful reading of and constructive suggestions/comments on our work. We address these below and indicate where changes were made in the revised manuscript.

Technical Comments:

Comment 1: *Consistency with experimental reports.*

This may be a typical criticism to many atomistic simulations. As far as I know that there is no experimental report on the CoCrFeNiPd HEA to date. Therefore, it is difficult to guarantee that the phenomena the authors reported can actually happen in reality. One suggestion is to cast an actual ingot of the HEA and make a very simple heat treatment to see whether the elemental partitioning like the authors reported actually happens or not. In the manuscript, the authors cited Z. Fu et al. (Ref. [32]) and other literature several times to support their results. However, the experimental conditions in the literature, such as chemical composition, processing conditions, and strain rate, are totally different. In addition, some of them seemed to be cited with inappropriate context. For instance, the authors cited Z. Fu et al. (Ref. [32]) to say that the nucleation of BCC phase could be easier than nucleating partial dislocations, but I don't think Z. Fu et al. argued such a thing in their paper. In my opinion, the authors should review the references they cited again to confirm the validity of their way to cite other literature.

Author reply: The reviewer is correct in noting that there are no experimental reports on CoCuFeNiPd HEA; comparison with such reports would, indeed, be ideal. This is why we have tried to present our study as comprehensively and as detailed as possible with references to relevant experiments and other calculations. For example, to ensure the accuracy of the interatomic potential used, we performed extensive, supporting density functional theory (DFT) calculations and validations (Table S1). Through such a comprehensive and detailed study, it was our intent to provide plausible and directly applicable guidelines for future experimental studies. We appreciate the reviewer's suggestion to cast an actual ingot of this HEA and perform a similar treatment and subsequent characterization. Currently, this is not a straightforward task for us in our laboratory and therefore we suggest leaving such experiments

to our future work. Due to the lack of the direct experimental support, we have tried to compare our simulation results with existing similar experimental observations and discussed their similarities and differences. As the reviewer correctly noted, such a criticism is applicable to most theoretical and modelling studies that try to explore issues for which experimental data is not yet available. We sincerely hope that the reviewer is amenable with this arrangement.

We thank the reviewer for comments regarding the accuracy of some references. Indeed, Z. Fu et al. [*Acta Mater.* **107**, 59 (2016)] did not observe the nucleation of a BCC phase in the $\text{Co}_{25}\text{Ni}_{25}\text{Fe}_{25}\text{Al}_{7.5}\text{Cu}_{17.5}$ HEA but did note that grain-boundary strengthening and dislocation strengthening are the principal mechanisms responsible for the high strength of this HEA. In a subsequent MD study by J. Li et al. [*Acta Mater.* **147**, 35 (2018)], it was shown that when dislocation nucleation and emission were suppressed, FCC to BCC phase transformations can occur. Therefore, we refer to J. Li et al.'s simulation results to support our deformation mechanisms: when dislocation nucleation and emission are suppressed, phase transformation from FCC to BCC can occur. Following the reviewer's suggestion, we rephrased the discussion on page 11 and cited this paper with the appropriate context. We have also carefully reviewed the other references and strived to present accurate and correct context.

To address the reviewer's concern, we have made the following changes in the revised manuscript:

Page 11: "A previous experimental study showed that the high strength of a $\text{CoCu}_{0.7}\text{FeNiAl}_{0.3}$ HEA³⁴ arose primarily from grain-boundary strengthening and dislocation-strengthening mechanisms. In a subsequent MD study on the same HEA²⁷, it was found that when dislocation nucleation and emission were suppressed, strain-induced phase transformations from the FCC to BCC phase occurred."

Page 21: "We note that phase transformation has been observed in an earlier simulation study²⁷."

Page 25: "These potential parameters have been employed in earlier HEA simulations with reasonable results^{27,46}."

Page 8: "Interestingly, Xu et al.³⁷ and Santodonato et al.³⁸ also observed Cu segregation in $\text{Al}_x\text{CoCrCuFeNi}$ HEAs, which they attributed to the addition of Al and the precipitation of a γ or B_2 phase."

Comment 2: *Universality of the phenomena.*

I think one of the most important parts of the manuscript is the formation of a composite-like structure due to SRO. Although the author focused on the HEA in the manuscript, I guess similar phenomena can occur not only in HEAs but simple binary or ternary systems. Thus, I wonder whether the authors found something special in the HEA or they wanted to say those are very common phenomena that we can observe in various alloy systems. Also, the authors employed a simple rule of mixture using DFT results to predict the local phase stability of the

HEA, and the results appeared to be consistent with the MD results. Why can we use such a very simple rule of mixture to predict atomistic properties? Is there any limitation of this methodology? The authors should address these points in the manuscript.

Author reply: The universality question raised by the reviewer is indeed interesting. Following the reviewer's suggestion, we added a brief discussion of the universality/non-universality of the phenomena. According to the equilibrium Gibbs phase rule $F = C - P + 2$, where F is the number of degrees of freedom, C is the number of components, and P is the number of phases. At a fixed temperature and stress ($F = 2$), only two phases ($P = 2$) are possible in a binary alloy ($C = 2$) and three phases ($P = 3$) in a ternary alloy ($C = 3$). Assuming that our HEA is in thermodynamic equilibrium, the observed FCC-preferred (FCCP), indifferent (IND), BCC-preferred (BCCP), and BCC domains after SRO is established correspond to $P = 4$ at fixed temperature and stress ($F = 2$). According to the Gibbs phase rule, there should require at least 4 components ($C = 4$) in the alloy. Thus, in an equilibrium condition, binary and ternary alloys should not exhibit such a composite microstructure. If the HEA is not in thermodynamic equilibrium, the Gibbs phase rule is not strictly followed, more phases are possible in the HEAs. Hence, the probability of formation of such a composite microstructure in HEAs with SRO will be much higher than that in simple binary or ternary alloy systems (where it can only form out of equilibrium).

The underlying origin for the enhancement of both strength and ductility is that the SRO-induced composite microstructure consists of three types of clusters: FCCP, IND, and BCCP. The IND clusters play the role of the matrix, the FCCP clusters serve as hard fillers that enhance the strength and the BCCP clusters act as soft fillers that increase the ductility. The formation of the composite microstructure is the large chemical-affinity disparity and high chemical-element exclusivity of HEA constituent elements [*Acta Mater.* **206**, 116638 (2021)]. Multiple elements are needed to form the different clusters/phases, e.g., an initial FCC phase in a random solid solution, and subsequent short-range ordered FCCP, BCCP, and IND clusters. Intrinsic FCC stabilizers are also essential to form the BCC phase during the strain-induced phase transformation from BCCP structures. These different requirements on the formation of such a composite microstructure require multiple elements.

Our simulations have shown that HEAs can be heterogeneous due to atomic segregation associated with SRO. During the SRO process, some elements prefer certain elemental neighbors while others are unfavored, these lead to local enrichment of the composition in spatially localized regions. Indeed, in the present work, we employed a rule of mixture to predict energy differences amongst the samples at 0, 2×10^6 and 4×10^6 iterations using data from DFT calculations; the predicted values were found to be consistent with MD results. In particular, our simulations show that SRO favors the following elemental pairs in the CoCuFeNiPd HEA: Co-Ni, Fe-Pd, and Cu-Cu. One of our goals was to estimate the cohesive energies of atomic pairs (nearest neighbors) from a surrogate structure that shares a cubic crystal lattice with the same coordination number/geometry as our FCC HEA. While many-

body interactions/correlations can be important, the standard lowest level approximation that admits SRO is the pair approximation. While there is no guarantee that such a pair approximation will suffice, the consistency of the cohesive energies from DFT calculations and MD results suggest that it is satisfactory in the present case.

In the revised manuscript, we have added the following discussions to emphasize these points.

Page 8: “Here we calculated the cohesive energies of (nearest neighbor) atomic pairs from a surrogate structure that shares a cubic crystal lattice with the same coordination number/geometry as our FCC HEA. The simple rule of mixture is the lowest level approximation of the many-body interactions/correlations in the HEA. In the present situation, this low level approximation is sufficiently accurate (based on the DFT and MD correspondence) to estimate the cohesive energies of the whole HEAs.”

Page 22: “While the Gibbs phase rule indicates that more phases can co-exist in equilibrium in an alloy with larger number of components, multiple phases can co-exist in binary and ternary alloys, but such microstructures are fundamentally unstable. We have shown that the SRO-induced composite microstructure consists of three types of clusters: “FCCP”, “IND”, and “BCCP” clusters. The formation of such a composite microstructure is associated with the SRO (previously identified⁴¹); the SRO is associated with a large chemical-affinity disparity and high chemical-element exclusivity amongst constituent chemical elements. Clearly, multiple elements are required to form these different atomic configurations and structures; e.g., the initial FCC phase in a random solid solution and the subsequent “FCCP”, “BCCP” and “IND” clusters that form as a result of SRO. Intrinsic FCC stabilizers are also essential to form the BCC phase during the strain-induced phase transformation from BCCP structures. These different requirements on the formation of such a composite microstructure require multiple elements. Hence, it is expected that the formation of such a composite microstructure is much more prevalent in HEAs than in simple binary or ternary alloys.”

Comment 3: Role of composite-like structure.

Many people say deformation-induced phase transformation can increase the work-hardenability of metallic materials, but, in my opinion, things may not be simple. I would like the authors to ask why the work-hardenability of the HEA increased in detail. One thing may be elastic interaction between FCC and BCC phase (or FCC phase parts having different chemical composition). I speculated that the parts having different elastic properties were dynamically formed and increased internal stress in the HEA due to the elastic interaction. Another thing can be the contribution of dislocations. It seemed to me there was some amount of plastic deformation in both FCC and BCC phases. I wonder how defects multiplied and how they were different depending on the crystal structure or stability of the parts. In addition, these two factors should have interactions, and there can be other contributions like the effect of transformation strain. The authors should comment on these aspects.

Author reply: We agree with the reviewer that the elastic interaction between FCC and BCC phase may play an important role in the work-hardenability of HEA via deformation-induced phase transformation. Figures S2 shows images of atomic configurations from a {001} cross-section of the CoCuFeNiPd HEA with 2×10^6 iterations at 300 K during tension at 300 K. We have added Fig. S7d to draw the atomic configurations of two domains in this sample at strain of 8% colored according to phase structures or stress distributions. Prior to dislocation nucleation, we observe that the elastic interaction between the FCC and BCC phases causes a stress concentration at the FCC and BCC interface, which in turn drives the FCC to BCC phase transformation. With increasing external strain, the elastic interactions lead to a larger stress concentration at the interface (Fig. S7d), inducing more phase transformation, and thus contributing to the work hardening (i.e., the stress increases with the phase transformation growing in the stress-strain curves of Fig. 2a). At a critical moment, the nucleation of dislocations occurs at the boundary between the FCC and BCC phases (Fig. S2), indicating that the elastic interactions are also the driving force for dislocation nucleation.

In this revised manuscript, we have added the following text and figure to discuss these aspects.

Page 23: “The elastic interaction between FCC and BCC phases plays an important role in the work-hardenability of HEA via the deformation-induced phase transformation. Figure S7d shows the atomic configurations of two domains in the 2M sample at strain of 8% colored according to phase structures or stress distributions. Prior to dislocation nucleation, we observe that the elastic interaction between the FCC and BCC phases causes a stress concentration at the FCC and BCC phase interface (Fig. S7d), which in turn drives the FCC to BCC phase transformation. With increasing external strain, the elastic interactions lead to a larger stress concentration at the interface, inducing more phase transformation, and thus contributing to the work-hardenability (i.e., the stress increases with the phase transformation growing in the stress-strain curves of Fig. 2a). At a critical moment, the nucleation of dislocations occurs at the boundary between the FCC and BCC phases (Fig. S2), indicating that the elastic interactions are also the driving force for dislocation nucleation.”

Fig. S2 Atomic configurations in a {001} cross-section of the CoCuFeNiPd HEA with 2×10^6 iterations at 300 K (the 2M sample) during tension at 300 K. Atomic configurations of the 2M sample at strains of **a** 8.3% (2M-300K-8.3%), **b** 8.4% (2M-300K-8.4%), **c** 8.5% (2M-300K-8.5%), **d** 8.6% (2M-300K-8.6%), **e** 8.7% (2M-300K-8.7%), and **f** 8.75% (2M-300K-8.75%) colored according to their common neighbor analysis phase structures.

Fig. S7 Atomic configurations of the CoCuFeNiPd HEA with 2×10^6 iterations at 300 K (the 2M sample) during loading and unloading at 300 K (2×10^8 s⁻¹ strain rate). Atomic configurations of the 2M sample at strains of **a** 2%, **b** 4%, and **c** 6% colored according to their stress distributions during loading. **d** Two domains in the 2M sample at strain of 8% colored according to phase structures or stress distributions. Atomic configurations of the 2M sample colored according to their **e** phase structures, and **f** strain distributions after unloading.

We would like to thank the reviewer again for her/his valuable comments and suggestions. We performed additional analyses and revised our manuscript to reflect both the spirit and substance of the reviewer's suggestions/questions; doing so has improved the clarity of our manuscript. We trust that these substantial changes will be satisfactory to the reviewer.

Response to Reviewer #2

Overall Comments: *This work demonstrates a new effect of the SRO in HEAs, i.e., SRO changes the local concentration, which makes the bcc/fcc-preferred region more stable than in the ideal random state. This helps the formation of pseudo-composite microstructures (FCCP, IND, BCCP) when loaded, leading to better strength and ductility simultaneously. This is an interesting work and contributes a new perspective to the recent hot topic, "SRO in HEAs". I recommend publishing it when the following questions are resolved in the coming revision.*

Author reply: We thank the reviewer for his careful reading of and positive assessment of our work. We consider each of the questions/suggestions from the reviewer below, as well as indicate how we revised the manuscript.

Technical Comments:

Comment 1: *Figure 2f shows that even at zero loadings, there are bcc regions in the 2M and 4M samples. Then how did you compute the WC-SRO parameter? How to define the neighboring shells? In Figure 1e-g, why the tables are not exactly symmetric?*

Author reply: Based on these questions, we revised our discussion of the calculation of WC-SRO parameters for increased clarity and included the definition of neighboring shells. We also corrected a very small rounding error in the WC-SRO tables.

Our calculations show that the atomic fraction of BCC-type structures in the samples after 2×10^6 (2M sample) and 4×10^6 (4M sample) iterations at zero loading is less than 3%, and these atoms are dispersed randomly through the whole domain. These BCC-type structures arise from local lattice distortion-induced instabilities. Interestingly, such transformation can be reversible under external influences, such as thermodynamic fluctuations or applied stress. To eliminate the influence of these atoms on the WC-SRO parameters, the samples are mapped to the initial structure with a perfect FCC structure (without lattice distortion; all neighbors on regular lattice sites). After mapping, the elemental types of 1st-nearest neighbors (as neighboring atoms) for each type of atom (as central atoms) are counted to calculate the WC-SRO parameters, using Equation 2.

The referee is, of course, correct in noting that the WC-SRO tables should be symmetric. The deviation from being exactly symmetric arose from rounding errors (deviations ≤ 0.01 in all cases in the 2 digits reported). We corrected these in the current manuscript.

In the revised manuscript, we have added the following text and revised the WC-SRO tables to clarify these points.

Page 12: “These BCC-type structures form as a result of local lattice distortion-induced instabilities and are dispersed randomly throughout the whole sample.”

Page 26: “To eliminate the influence of lattice distortions in the calculation of the WCPs, the atomic site occupancies are mapped to the corresponding perfect FCC structure (no lattice distortion; all atoms on regular lattice sites). The WC_{nm} calculations were performed on this lattice by counting the elemental types of 1st-nearest neighbors.”

Fig. 1 Warren-Cowley parameters at 0 (e), 2×10^6 (f), and 4×10^6 (g) iterations.

Comment 2: One of the key concepts in the draft is that SRO produces a pseudo-composite microstructure, leading to better mechanical properties as compared to the random state. But I think we need to be more careful about this statement. As shown in Figure 2b-c, even the ideal random state (sample 0) can have the composite microstructure at a strain of 6%, with the BCC fraction of ~12%. Therefore, it seems that the effect of SRO is to (1) make the BCCP domain easier to happen, (2) increase the fraction of the BCC phase. Then my question is why more BCCP is better for mechanical properties, as compared to the random state? Is there any optimal fraction of BCCP? It seems insufficient to only discuss the composite-based mechanism itself, as it applies in both random and SRO alloys. It would be better to add more analysis and discussions about the effect of SRO on the alloy mechanical properties, which is the reason why we shall use SRO to improve the alloys, instead of using just a random alloy.

Author reply: Following the reviewer’s suggestions, we performed additional simulations with more iterations starting from the 4M sample. In order to obtain samples with a higher degree of SRO, we calculated the sum of squares for the Warren-Cowley parameters for each MC attempt, and accepted those attempts that increased the sum of squares. After 10^4 successful MC iterations, a sample (labeled as the 4M+ sample) with a similar potential energy (Fig. S5a) but (slightly) larger FCCP and BCCP fractions (Fig. S5b) is obtained. The Warren-Cowley parameters of 4M+ sample (Fig. S5c) clearly exhibit a higher degree of SRO, as compared with the 4M sample (Fig. 1g). The stress-strain curves for the 0, 2M, 4M, and 4M+ samples under tension at 300 K are plotted in Fig. S5d; the 2M, 4M, and 4M+ sample have similar ultimate tensile strengths (that of 4M sample is slightly larger). Our detailed analysis of the atomic configurations of the 4M+ sample at 9% and 10% strains (Fig. S5e) shows that the deformation mechanism of the 4M+ sample is the same as that of other samples. Therefore, these new simulation results demonstrate that there are optimal fractions for FCCP and BCCP

structures. Our results demonstrate, in agreement with the reviewer, that SRO increases the BCCP fraction and makes the BCCP domain easier to transform to BCC phase. This is why the SRO alloys (the 2M, 4M, and 4M+ samples) have better mechanical properties than the random alloy. Further increase in the BCCP fraction leads to more BCC phase formed, inducing more phase boundaries in the 4M+ sample than in the 4M sample. Since dislocation nucleation occurs at the FCC/BCC phase boundaries (Fig. S2), this implies more/easier dislocation nucleation in the 4M+ sample than in the 4M sample. This explains why there is an optimal BCCP fraction structure.

In the revised manuscript, we have added the following figure and text to emphasize these points.

Fig. S5 Potential energies, fractions of atoms with different phase preferences, Warren-Cowley parameters, stress-strain curves, and atomic configurations of the CoCuFeNiPd HEA for different iterations at 300K. a Variation of the potential energy with iteration. **b** Fractions of atoms with different phase preferences for the HEA with 0 (0 sample), 2×10^6 (2M sample), 4×10^6 (4M sample), and $4 \times 10^6 + 1 \times 10^4$ (4M+ sample) iterations. **c** Warren-Cowley parameters for the 4M+ sample. **d** Uniaxial tensile stress-strain curves of the 0, 2M, 4M, and 4M+ samples during tension at 300K. **e** Atomic configurations of the 4M+ sample at 9% and 10% strains colored according to the common neighbor analysis phase structures.

Page 16: “We performed additional simulations with more iterations, starting from the 4M sample to further enhance the SRO. The results are shown in Fig. S5 and discussed in

Supplementary Information, indicating there is an optimal “FCCP” and “BCCP” fraction structure.”

Page S3: “We performed additional simulations with more iterations, starting from the 4M sample to further enhance the SRO. We calculated the sum of squares of the WCPs for each MC attempt and only retained the changes that increased the sum of squares. After 10^4 successful MC iterations, we obtain sample 4M+ which has a similar potential energy (Fig. S5a) but larger “FCCP” and “BCCP” fractions (Fig. S5b). The WCPs of 4M+ sample (Fig. S5c) show a higher degree of SRO than for the 4M sample (Fig. 1g). The tensile stress-strain curves for the 0, 2M, 4M, and 4M+ samples (Fig. S5d) show that the largest ultimate tensile strength is obtained for the 4M sample. Detailed analysis of the 4M+ atomic configurations at 9% and 10% strains (Fig. S5e) shows that the deformation mechanism of the 4M+ sample is the same as that of other samples. Therefore, these simulation results demonstrate that there are optimal fractions for “FCCP” and “BCCP” structures. Our results demonstrate that SRO increases the “BCCP” fraction and makes the “BCCP” domain easier to transform to BCC phase. This is why the SRO alloys (the 2M, 4M, and 4M+ samples) have better mechanical properties than the random alloy. Further increase in the “BCCP” fraction leads to more BCC phase formed, inducing more phase boundaries in the 4M+ sample than in the 4M sample. Since dislocation nucleation occurs at the FCC/BCC phase boundaries (Fig. S2), this implies more/easier dislocation nucleation in the 4M+ sample than in the 4M sample. This explains why there is an optimal BCCP fraction structure.”

Comment 3: *The fcc alloys usually have lower strength and more ductility as compared to the bcc alloys. See reference: <https://doi.org/10.1016/j.actamat.2020.01.062>. However, it is quite the opposite in the FCCP and BCCP phases, as shown in Figure 5a. The FCCP phase is the hardening domain and Ni-Co-rich, but the NiCo alloy has a very low yield strength as shown in the reference above. Is there any discussion for that?*

Author reply: In our HEA, both FCCP and BCCP clusters are in the FCC phase. In the 2M sample, FCCPs contain 29at% Co, 18at% Cu, 4at% Fe, 48at% Ni, and 1at% Pd, while BCCPs contain 15at% Co, 6at% Cu, 31at% Fe, 4at% Ni, and 44at% Pd (on average). Following the reviewer’s suggestion, we performed additional tensile test simulations on a pair of samples with the same elemental concentrations as the FCCP and BCCP clusters in the 2M sample. The stress-strain curves of these two samples (labelled FCCP and BCCP) are shown in Fig. S12a (along with the results from the 2M sample), further verifying that FCCP is the hardening domain. Atomic configurations of the FCCP and BCCP samples at different strains are drawn in Figs. S12b and c, respectively, demonstrating that the deformation mechanism is dislocation slip for FCCP and phase transformation for BCCP. These results further confirm that BCCP clusters are metastable within the FCC phase and they transform to BCC phases during tensile deformation. Such phase transformation is responsible for the observed plasticity and ductility.

NiCo in the reference [*Acta Mater.* **188**, 486–491 (2020)] is a random alloy (a solid solution) with pre-existing dislocation. Therefore, the yield strength is dominated by solute-misfit volumes (based on their theoretical framework). Since NiCo has near-zero misfit volume, the NiCo random solid solution exhibits a very low yield strength. In our simulations, the five element FCCP clusters (29 Co, 18 Cu, 4 Fe, 48 Ni, 1 Pd (at%)) exhibit SRO (obviously) not present in random NiCo alloy. Besides, FCCP clusters do not have pre-existing dislocation. Hence, it is no surprise that the FCCP clusters in the HEAs exhibit quite different mechanical properties from random NiCo alloys.

In the revised manuscript, we have added the following figure and text (with references):

Fig. S12 Tensile stress-strain curves and atomic configurations of the samples with the same elemental concentrations as “FCCP” and “BCCP” clusters in the 2M sample at 300K (labelled as “FCCP” and “BCCP” samples). **a** Stress-strain curves of the “FCCP” and “BCCP” samples, along with the results of the 2M sample. Atomic configurations of **b** the “FCCP” sample and **c** the “BCCP” sample at different strains colored according to their common neighbor analysis phase structures.

Page 22: “Examination of the two samples with the same elemental concentrations as the “FCCP” and “BCCP” clusters (Fig. S12a) verifies that “FCCP” is the hardening domain. Random FCC NiCo alloys with pre-existing dislocation have yield strengths dominated by solute-misfit⁴². However, NiCo has a near-zero misfit such that random NiCo alloys exhibit a very low yield strength⁴². In our simulations, the five-element “FCCP” clusters (29 Co, 18 Cu, 4 Fe, 48 Ni, 1 Pd (at%)) exhibit SRO (obviously) not present in random NiCo alloys. Besides, “FCCP” clusters do not have pre-existing dislocation. Hence, it is no surprise that the “FCCP” clusters in the HEAs exhibit quite different mechanical properties from random NiCo alloys.”

Page S7: “In our HEA, both “FCCP” and “BCCP” clusters are in an FCC phase. In the 2M sample, “FCCP” clusters have an average composition of 28 Co, 18 Cu, 4 Fe, 48 Ni, 2 Pd (at%), while the average composition of “BCCP” clusters is 15 Co, 6 Cu, 31 Fe, 4 Ni, 44 Pd (at%). Two samples with the same elemental concentrations as the “FCCP” and “BCCP” clusters were constructed to perform tensile tests. Examination of the stress-strain curves for these two samples (see Fig. S12a) verifies that “FCCP” is the hardening domain. Inspection of the atomic

configurations of the “FCCP” and “BCCP” samples at different strains (Figs. S12b and c) demonstrates that the dominant deformation mechanism is dislocation slip in “FCCP” and phase transformation in “BCCP”. These results further confirm that “BCCP” clusters are metastable within the FCC phase and transform to BCC phase during tensile deformation. Such phase transformation contributes to both the plasticity and ductility of the material. Random FCC NiCo alloys with pre-existing dislocation have yield strengths dominated by solute-misfit^{S3}. However, NiCo has a near-zero misfit such that random NiCo alloys exhibit a very low yield strength^{S3}. In our simulations, the five-element “FCCP” clusters (29 Co, 18 Cu, 4 Fe, 48 Ni, 1 Pd (at%)) exhibit SRO (obviously) not present in random NiCo alloys. Besides, “FCCP” clusters do not have pre-existing dislocation. Hence, it is no surprise that the “FCCP” clusters in the HEAs exhibit quite different mechanical properties from random NiCo alloys.”

Comment 4: *The MC simulation was conducted at 300 K. In practice, such a low annealing temperature would prohibit the kinetic of diffusion. It would be better to add some discussion about the feasibility of the composite-based mechanism proposed in this work.*

Author reply: In the revised manuscript, we added discussions on the effect of annealing temperature on composite microstructure. To aid these discussions, we performed additional MC/MD simulations at 1,200 K. Not surprisingly, the same SRO trends observed at 300 K (Fig. 1) are also seen at 1,200 K (Fig. S10). However, the stronger entropic effects at higher temperature greatly reduce enthalpic (bond energy) effects. As a result, the SRO at the higher temperature is reduced. This is consistent with the observation (Fig. S10c) that the energy per atom decreases less at 1,200 K, as compared with that at 300 K.

The tensile stress-strain curves for the HEAs with 0, 2×10^6 , and 4×10^6 iterations at 1,200 K (0, 1,200K-2M, and 1,200K-4M samples) under tension at 300 K and 1,200 K (a strain rate of $2 \times 10^8 \text{ s}^{-1}$) are shown in Fig. S11a. These data indicate that the ultimate stress and associated ultimate strain for the 1,200K-2M and 1,200K-4M samples (iterated at 1,200 K) are slightly lower, as compared with the 0, 2M and 4M samples (iterated at 300 K), suggesting that the SRO in the HEAs after iterations at 1,200 K slightly reduces the strength and ductility. Examinations of the atomic configurations in Figs. S11b and c demonstrate that the deformation mechanism remains unchanged (phase transformation from FCC to BCC/amorphous structures prior to the ultimate stress and dislocation slip afterwards).

The distributions of number fractions of atoms with $\Delta E_{\text{FCC-BCC}}$ for the 1,200K-2M, and 1,200K-4M samples are plotted in Fig. S11d, as compared with those at 300 K (the 2M and 4M samples). The curve shapes of the 1,200K-2M and 1,200K-4M samples are quite different from those for the 2M and 4M samples in Fig. S11d. There exist two plateaus at $\sim -3 \times 10^{-2} \text{ eV}$ and $\sim 1 \times 10^{-2} \text{ eV}$, respectively, for the 2M and 4M samples, indicating the formation of three categories of clusters. However, the curve shapes of the 1,200K-2M and 1,200K-4M samples imply that there is no such cluster formation. Besides, the atomic configurations colored according to their

phase stability for the 1,200K-2M and 1,200K-4M samples shown in Figs. S11e and f further confirm that their atomic configurations are different from that of the 2M sample shown in Fig. 3e, i.e., without FCCP and BCCP serving as hard and soft fillers in the matrix (IND) of the HEA to form a composite microstructure. This is why the 1,200K-2M and 1,200K-4M samples do not exhibit better mechanical properties than the 0, 2M, and 4M samples.

In the revised manuscript, we have added the following figures and text to discuss the feasibility of the composite-based mechanism regarding to the temperature effect.

Fig. S10 Warren-Cowley parameters and potential energies of the CoCuFeNiPd HEA as a function of iterations. Warren-Cowley parameters of the HEA after **a** 2×10^6 and **b** 4×10^6 iterations at 1,200 K. **c** Variation of potential energies with iterations at 300 K and 1,200 K.

Fig. S11 Dependence of the stress-strain curves, atomic configurations, and phase stabilities of the CoCuFeNiPd HEA on number of iterations at 1,200 K. **a** Uniaxial tensile stress-strain curves for the HEAs with 0, 2×10^6 , and 4×10^6 iterations at 1,200 K (the 0, 1,200K-2M, and 1,200K-4M samples)

under tension at 300 K and 1,200 K. Atomic configurations of the 1,200K-2M sample under tension at 300 K at **b** 5.6% (1,200K-2M-300K-5.6%) and **c** 6.6% (1,200K-2M-300K-6.6%) strain, colored according to their common neighbor analysis phase structures (marked in **a**). **d** Distribution of atoms according to their phase stability $\Delta E_{\text{FCC-BCC}}$ for the 1,200K-2M and 1,200K-4M samples, as compared with the 2M and 4M samples. Atomic configurations of the **e** 1,200K-2M sample, and **f** 1,200K-4M sample colored according to phase stability ($\Delta E_{\text{FCC-BCC}}$).

Page 17: “We also explored the effect of annealing temperature on SRO and mechanical properties of HEAs; associated results are shown in Figs. S10 and S11 and discussed in Supplementary Information.”

Page S6: “To investigate the effect of annealing temperature on segregation and SRO, we performed MC and MD simulations on the CoCuFeNiPd HEA at 1,200 K. Not surprisingly, the same trends of SRO observed at 300 K (Fig. 1e to g) are also seen at 1,200 K (Fig. S10a and b). However, the stronger entropic effects at higher temperature greatly reduce enthalpic (bond energy) effects. As a result, the SRO at the higher temperature is reduced. This is consistent with the observation (Fig. S10c) that the energy per atom decreases less at 1,200 K, as compared with that at 300 K.

The tensile stress-strain curves for the HEAs with 0, 2×10^6 , and 4×10^6 iterations at 1,200 K (0, 1,200K-2M, and 1,200K-4M samples) under tension at 300 K and 1,200 K (a strain rate of $2 \times 10^8 \text{ s}^{-1}$) are shown in Fig. S11a. These data indicate that the ultimate stress and associated ultimate strain for the 1,200K-2M and 1,200K-4M samples (iterated at 1,200 K) are slightly lower, as compared with the 0, 2M, and 4M samples (iterated at 300 K), suggesting that the SRO in the HEAs after iterations at 1,200 K slightly reduces the strength and ductility. Examinations of the atomic configurations in Figs. S11b and c demonstrate that the deformation mechanism remains unchanged (phase transformation from FCC to BCC/amorphous structures prior to the ultimate stress and dislocation slip afterwards).

The distributions of number fractions of atoms with $\Delta E_{\text{FCC-BCC}}$ for the 1,200K-2M, and 1,200K-4M samples are plotted in Fig. S11d, as compared with those at 300 K (the 2M and 4M samples). The curve shapes of the 1,200K-2M and 1,200K-4M samples are quite different from those for the 2M and 4M samples. There exist two plateaus at $\sim -3 \times 10^{-2} \text{ eV}$ and $\sim 1 \times 10^{-2} \text{ eV}$, respectively, for the 2M and 4M samples, indicating the formation of three categories of clusters (“FCCP”, “IND”, and “BCCP” clusters). However, the curve shapes of the 1,200K-2M and 1,200K-4M samples imply that there is no such cluster formation. Besides, the atomic configurations colored according to their phase stability for the 1,200K-2M and 1,200K-4M samples shown in Figs. S11e and f further confirm that their atomic configurations are different from that of the 2M sample presented in Fig. 3e, i.e., without “FCCP” and “BCCP” serving as hard and soft fillers in the matrix (“IND”) of the HEA to form a composite microstructure. This is why the 1,200K-2M and 1,200K-4M samples do not exhibit better mechanical properties than the 0, 2M, and 4M samples.”

We would like to thank the reviewer again for his valuable comments and suggestions. We have performed additional calculations and revised our manuscript to reflect both the spirit and substance of the reviewer's comments and suggestions.

Response to Reviewer #3

Overall Comments: *In this paper the authors report on a study of the atomic structures and mechanical behavior of a CoCuFeNiPd high entropy alloy (HEA). Unlike other well studied approaches that have been used to promote strength-ductility synergy in HEAs, such as precipitation, recrystallization, tuning the chemistry, and/or adding elements such as oxygen nitrogen and others authors of this study propose a strategy that involves short range order (SRO). In principle, SRO can increase the resistance to dislocation glide thereby yielding an increase in strength. This can be particularly effective when the microstructure contains a mixture of phases. Interestingly, the authors use an approach that involves a combination of the Monte Carlo method together with molecular dynamics and density functional theory. They use atomic simulations as a tool to understand both segregation and short range order(SRO) in this HEA. The paper is very well written and provides insight into phenomena that would be very difficult to assess experimentally. Like with all simulations one always wonders what would happen if for example deformation would occur at realistic strain rates. There are a few points that should be addressed prior to publication.*

Author reply: We thank the reviewer for her/his careful and supportive reading of our manuscript. We have carefully considered the reviewer's input and revised our manuscript in line with the suggestions/comments, as outlined below.

Technical Comments:

Comment 1: *In the Introduction, it is surprising that the current authors did not mention the two important references related to SRO in the complex alloy field. "Zhang, F. X. et al. Local structure and short-range order in a NiCoCr solid solution alloy. Phys. Rev. Lett. 118, 205501 (2017)." "Zhang, R.P. et al. Short-range order and its impact on the CrCoNi medium-entropy alloy. Nature volume 581, pages283–287(2020)."*

Author reply: We appreciate the reviewer pointing out our omission of these two important papers. We aimed to keep a balance between comprehensiveness and brevity in the Introduction. Therefore, we mainly focused on short-range ordering (SRO) in high-entropy alloys (HEAs) and, less so, on medium-entropy alloys (MEAs), such as NiCoCr. In our revision, we expanded our review to briefly discuss MEAs and these pertinent references [*Phys. Rev. Lett.* **118**, 205501 (2017); *Nature* **581**, 283–287 (2020)].

Page 4: "SRO also occurs in medium-entropy alloys (MEAs). Atomic structure of a CoCrNi MEA indicates that Cr favors Ni and Co neighbors, reducing the electrical and thermal conductivities²⁴ and increasing the stacking-fault energies and hardness²⁵."

Comment 2: In Page 7 (line 133 and 134), for Cu segregation in $Al_xCoCrCuFeNi$ HEAs, the addition of Al contributes to the precipitation of γ' in Reference 35 or B2 in Reference 36, which causes the Cu segregation in the matrix. I think the conclusion in this part is not convincing.

Author reply: Duly noted. We revised the manuscript as follows.

Page 8: “Interestingly, Xu et al.³⁷ and Santodonato et al.³⁸ also observed Cu segregation in $Al_xCoCrCuFeNi$ HEAs, which they attributed to the addition of Al and the precipitation of a γ or B₂ phase.”

Comment 3: In Page 9 (line 160), the strain rate for the tensile simulation is as high as $2 \times 10^8/s$, it seems this work belongs to ultra-high strain rate region. Therefore, I am wondering if the current simulated results can represent the mechanical properties of alloys under normal tensile test (10^{-4} - $10^{-3}/s$).

Author reply: Of course, the reviewer is correct in noting that the deformation rates employed $2 \times 10^8 \text{ s}^{-1}$ represent an ultra-high strain rate. Such an ultra-high strain rate is more typically of extreme loading conditions, for example, laser shock loading [*Phys. Rev. Lett.* **111**, 065501 (2013); *Phys. Rev. Lett.* **95**, 075501 (2005)]. Currently, it is not possible to perform MD simulations at the more conventional experimental strain rates (10^{-4} - 10^{-3} s^{-1}) due to the extreme computational cost. To address this issue, we follow the interesting work by Li et al. [*Nature* **464**, 877 (2010)], in which a theory coupling strength (σ) and strain rate (ε) was formulated by consideration of dislocation nucleation kinetics:

$$\sigma = \frac{\Delta U}{SV^*} - \frac{k_B T}{SV^*} \ln \frac{d v_D}{\lambda \varepsilon}$$

where ΔU is the activation energy, S is a factor representing the local stress concentration and geometry, V^* is the activation volume, $k_B T$ is a thermal energy, v_D is the Debye frequency, d and λ are microstructure parameters, respectively. Following this formulation, we simplify the relation as follows:

$$\sigma = C + K \ln \varepsilon$$

where C and K are constants, which we determine from the simulation ultimate stresses (of the 0 and 4M samples) at $2 \times 10^8 \text{ s}^{-1}$ and $2 \times 10^7 \text{ s}^{-1}$ strain rates. Using this equation and the values of C and K , we estimate the ultimate stresses of these two samples for a strain rate of $2 \times 10^{-3} \text{ s}^{-1}$; i.e., the ultimate stress for the 4M and 0 samples to be 3.92 GPa and 3.13 GPa, respectively. These results shows that the sample with SRO has a higher ultimate stress than the sample without SRO even at a low strain rate of $2 \times 10^{-3} \text{ s}^{-1}$. Therefore, our simulation results at ultra-high strain rates can also be used to predict the trend of mechanical properties of alloys under low-strain-rate tensile tests.

In the revised manuscript, we have added the following text to discuss the effect of strain rate.

Page 12: “For the low experimental strain rates (10^{-4} - 10^{-3} s $^{-1}$), we employed Li et al.’s theory⁴⁰ for further exploration and discussed the results in Supplementary Information.”

Page 27: “Strain rates of 2×10^8 and 2×10^7 s $^{-1}$ have been achieved, e.g., under laser shocking^{49,50}.”

Page S2: “Li et al.^{S1} formulated a theory connecting strength (σ) and strain rate ($\dot{\epsilon}$) in a dislocation nucleation kinetics-based model:

$$\sigma = \frac{\Delta U}{SV^*} - \frac{k_B T}{SV^*} \ln \frac{d\nu_D}{\lambda \dot{\epsilon}} \quad (S1)$$

where ΔU is the activation energy, S is a factor representing the local stress concentration and geometry, V^* is the activation volume, $k_B T$ is the thermal energy, ν_D is the Debye frequency, d and λ are microstructure parameters. Following their approach, we simplify this relation as:

$$\sigma = C + K \ln \dot{\epsilon} \quad (S2)$$

where C and K are constants, which we determine from the simulation ultimate stresses (of the 0 and 4M samples) at 2×10^8 s $^{-1}$ and 2×10^7 s $^{-1}$ strain rates. Using Eq. (S2) and the values of C and K , we estimate the ultimate stresses of these two samples for a strain rate of 2×10^{-3} s $^{-1}$; i.e., the ultimate stress for the 4M and 0 samples to be 3.92 GPa and 3.13 GPa, respectively. These results shows that the sample with SRO has a higher ultimate stress than the sample without SRO even at a low strain rate. Therefore, our simulation results at ultra-high strain rates suggests a similar trend in mechanical properties at low strain rates.”

Comment 4: *Also about this ultra-high strain rate, the recent work by Zhao et al. (Sci. Adv. 2021; 7: eabb3108) shows that even under strain rate of 6×10^5 /s, an amorphous phase would appear. I am wondering what phase would be formed when the strain rate is 100 higher. Is it possible to form amorphous phase in this sample instead of BCC or HCP at such high strain rates?*

Author reply: In Zhao et al.’s work [Sci. Adv. 7, eabb3108 (2021)], the amorphous phases were observed in a CoCrFeNiMn HEA at a strain rate of 6×10^5 s $^{-1}$, which they attributed to the high defect density from the initial pre-processing conditions. However, the initial structure of our CoCuFeNiPd HEA is perfect (i.e., contains no defects). Hence, it is expected that the nucleation of an amorphous structure in our perfect HEA material would be difficult. Indeed, in our simulations under the strain rates of 2×10^8 and 2×10^7 s $^{-1}$, we do not observe amorphous phases.

In the revised manuscript, we have added the following text to emphasize this point.

Page S2: “We note that Zhao et al.^{S2} observed the formation of amorphous phases in a CoCrFeNiMn HEA under a strain rate of 6×10^5 s $^{-1}$. These amorphous phases were attributed to the high defect density from the initial pre-processing conditions^{S2}. However, the initial structure of our CoCuFeNiPd HEA is perfect (i.e., contains no defects). As a result, the

nucleation of amorphous phases is suppressed, even at high strain rates of 2×10^8 and 2×10^7 s^{-1} .”

Comment 5: *In Fig. 2a, in the ss curves, it seems that the elastic strain is over 6%, which is similar to superelastic behavior. Since the theoretical maximum superelastic strain is closely related to the lattice parameter of austenite and martensite phases, what is the lattice parameters of the FCC matrix, strain or stress-induced BCC and HCP phases.*

Author reply: The average lattice parameters of the FCC phase for the 0, 2M, and 4M samples at a 6% strain at 300 K (no dislocations) are found to be 3.685 Å, 3.662 Å, and 3.642 Å, respectively; i.e., the average lattice parameter decreases during SRO formation. The BCC phase, on the other hand, shows average lattice parameters of 2.959 Å, 2.957 Å, and 2.965 Å for the 0, 2M, and 4M samples, respectively; i.e., only small, statistically insignificant change during SRO formation.

In the revised manuscript, we have added the following text to discuss the lattice parameters.

Page S8: “The average lattice parameters of the FCC phase for the 0, 2M, and 4M samples at a 6% strain at 300 K (no dislocations) are found to be 3.685 Å, 3.662 Å, and 3.642 Å, respectively; i.e., the average lattice parameter decreases during SRO formation. The BCC phase, on the other hand, shows average lattice parameters of 2.959 Å, 2.957 Å, and 2.965 Å for the 0, 2M, and 4M samples, respectively; i.e., only small, statistically insignificant change during SRO formation.”

Comment 6: *In Fig. 2f, for 2M and 4M sample, it seems that 5% of BCC phase already exists before tensile tests. Why?*

Author reply: Figure 2f shows that, for the 2M and 4M samples, there is slightly less than 3% BCC-type structures present prior to tensile testing. These BCC-type structures form as a result of local lattice distortion-induced instabilities and are dispersed randomly throughout the whole system. These BCC clusters revert to FCC with thermodynamic fluctuations or applied stress.

In the revised manuscript, we have added the following text to make this point clearer.

Page 12: “These BCC-type structures form as a result of local lattice distortion-induced instabilities and are dispersed randomly throughout the whole sample.”

Comment 7: *In Fig. 2, it seems that the strength decreases with increasing amount of HCP phase. Why?*

Author reply: The HCP structures plotted in Fig. 2 are simply stacking faults arising from the nucleation and propagation of partial dislocations (Fig. S2). Therefore, the nucleation of the HCP structure is associated with a decrease in stress (plasticity). After nucleation, further

increase in the HCP structure fraction is associated with dislocation propagation. The propagation of partial dislocations occurs in relatively lower stress until blocked by phase boundaries or other dislocations (Fig. S2f), requiring an increase in stress for further movement, i.e., strain hardening.

In the revised manuscript, we have added the following text to elucidate these trends.

Page 10: “The HCP structure regions seen in Figs. 2, S1, and S2 are stacking faults associated with partial dislocation nucleation (Fig. S2b), which corresponds to a local (plastic) release of stress. The propagation of these partial dislocations occurs in relatively lower stress until blocked by phase boundaries or other dislocations (Fig. S2f), requiring an increase in stress for further motion, i.e., strain hardening.”

Fig. S2 Atomic configurations in a $\{001\}$ cross-section of the CoCuFeNiPd HEA with 2×10^6 iterations at 300K (the 2M sample) during tension at 300K. Atomic configurations of the 2M sample at strains of **a** 8.3% (2M-300K-8.3%), **b** 8.4% (2M-300K-8.4%), **c** 8.5% (2M-300K-8.5%), **d** 8.6% (2M-300K-8.6%), **e** 8.7% (2M-300K-8.7%), and **f** 8.75% (2M-300K-8.75%) colored according to their common neighbor analysis phase structures.

Comment 8: In Page 11 (line 196 and 197), the current authors claimed that strain-induced phase transformations dominate the deformation of the HEA, I would like to know if the current authors did any calculation for the critical stress for phase transformation.

Author reply: We performed additional analyses of the critical stress for phase transformation.

The variations of phase transformation and stress as a function of strain for the 0 and 4M samples for a 300 K tensile test are shown in Fig. S6a. The results indicate that the critical strain for phase transformation in the 4M and 0 samples are ~2.6% (continuous blue line) and 4% (dashed blue line), corresponding to critical stresses of 1.6 GPa (continuous red line) and 2.3 GPa (dashed red line), respectively. Atomic configurations of the 0 and 4M samples at the critical strain and stress for phase transformation are shown in Figs. S6b and c. The sample with greater SRO (4M) has lower critical stress for phase transformation due to the formation of more BCCP clusters.

In the revised manuscript, we have added the following figure and text to discuss the critical stress for phase transformation.

Fig. S6 Fraction of phase transformation, stress vs strain curves and atomic configurations of the CoCuFeNiPd HEA of the random (0) and 4×10^6 SRO (4M) samples tested in tension at 300 K. a Variations of phase transformation fraction and stress with strain for the 0 and 4M samples during tension at 300 K. Atomic configurations of **b** the 0 sample at a 4% strain (0-4%), and **c** the 4M sample at a 2.6% strain (4M-2.6%) colored according to their common neighbor analysis phase structures.

Page 16: “The critical stress required for phase transformation in the sample with/without SRO is also analyzed and presented in Supplementary Information (Fig. S6).”

Page S4: “The variations of phase transformation and stress as a function of strain for the 0 and 4M samples for a 300 K tensile test are shown in Fig. S6a. The results indicate that the critical strain for phase transformation in the 4M and 0 samples are ~2.6% (continuous blue line) and 4% (dashed blue line), corresponding to critical stresses of 1.6 GPa (continuous red line) and 2.3 GPa (dashed red line), respectively. Atomic configurations of the 0 and 4M samples at the critical strain and stress for phase transformation are shown in Figs. S6b and c. The sample with greater SRO (4M) has lower critical stress for phase transformation due to the formation of more “BCCP” clusters.”

Comment 9: In Fig.3, for the SRO and phase transformation in the current HEA, it seems that the Ni segregation helps to tune the martensite start temperature, which results in the

heterogeneity of FCC-BCC transformation in the current samples. So the key to adjusting SRO and related phase transformation in the current HEA is the local concentration of Ni?

Author reply: The FCC-BCC transformation occurs in BCCP clusters, which are enriched in Fe and Pd. Therefore, the key to adjusting SRO and related phase transformation in the CoCuFeNiPd HEA is to tune the BCCP cluster concentrations of Fe and Pd. The change in the local concentration of Ni may originate from FCCP clusters (primarily Co and Ni) or IND clusters (consisting of all five elements). If the change in the local concentration of Ni occurs in the FCCP clusters, it may have a minor effect on the BCCP clusters and phase transformation. However, if the change comes from the IND clusters, it may introduce more BCCP clusters. Therefore, it may also affect the related phase transformation.

In the revised manuscript, we have added the following text to emphasize these points.

Page S3: “The FCC to BCC transformation occurs in “BCCP” clusters, which are enriched in Fe and Pd. Therefore, the key to adjusting SRO and related phase transformation in the CoCuFeNiPd HEA is to tune the “BCCP” cluster Fe and Pd concentrations. The change in the local concentration of Ni may originate from “FCCP” clusters (primarily Co and Ni) or “IND” clusters (consisting of all five elements). If the change in the local concentration of Ni occurs in the “FCCP” clusters, it may have a minor effect on the “BCCP” clusters and phase transformation. However, if the change comes from the “IND” clusters, it may introduce more “BCCP” clusters. Therefore, it may also affect the related phase transformation.”

***Comment 10:** In Fig. 4, for the SRO with clusters of FCCP, IND and BCCP, it seems this kind of SRO is very similar to one with nano-scale precipitates, even in the element concentration. How can one clearly define SRO clusters and precipitates (3-5 nm) in the current HEAs?*

Author reply: We added a more explicit definition on SRO clusters. In our HEA, the SRO clusters have similar elemental concentrations and the same crystal structures as the HEA matrix. For example, FCCP, IND, and BCCP are SRO clusters in the HEA that have similar elemental concentrations and the same FCC crystal structure. However, the crystal structure and composition of a nanoscale precipitate in an HEA are well defined and are generally different from those of the HEA matrix. Since the atomic segregation in our HEA is related to SRO, the FCCP, IND, and BCCP have not reached a level of maturity to form well-defined chemical compositions, and thus do not “qualify” as precipitates.

In the revised manuscript, we have added the following text to distinguish SRO clusters and nano-scale precipitates.

Page 19: “In our HEA, SRO clusters have similar elemental concentrations and the same crystal structures as the HEA matrix. For example, “FCCP”, “IND”, and “BCCP” are SRO clusters that have similar elemental concentrations and the same FCC crystal structure. On the other hand, the crystal structure and chemical composition of a nanoscale precipitate in a HEA are

well defined and generally differ from those of the HEA matrix. Hence, we do not consider the SRO clusters as precipitates.”

Comment 11: In Fig. 5c and 5d, for the loading-unloading ss curves, this HEA undergoes reversible transformation (strain up to 8%). I am curious about the type of BCC phase? Is it thermoelastic?

Author reply: Indeed, some of the BCC phase regions undergo reversible transformations to the FCC structure. Since this reversible phase transformation occurs during unloading, we believe that the reverse phase transformation is a strain-induced phase transformation. Since the simulations were isothermal, this phase transformation is not a thermoelastic process.

In the revised manuscript, we have added the following text to discuss the type of the BCC phase.

Page 17: “It is interesting to note that some of the BCC domains undergo reversible transformation to the FCC phase upon unloading. Since the simulations are isothermal, this phase transformation is strain-induced rather than thermoelastic.”

Comment 12: In Table 1, for the first four types of HEAs, they are all polycrystalline and grain boundaries play an important role in tailoring their mechanical properties. However, the current HEA is single crystal, and grain boundaries are not taken into account. Based on their simulations and calculations, it seems that more severe element segregation would take place along grain boundaries. I would like to know if the SRO would still improve mechanical properties of this HEA when the grain boundaries are considered.

Author reply: This is an interesting point. Following the reviewer’s suggestion, we performed additional simulations with grain boundaries considered. A model of two grains in the same size but with different orientations was constructed by aligning them along [010] direction (Fig. S8c), where one grain is identical with the previous single-crystal one (i.e., 5-nm cube with [100], [010] and [001] directions as shown in Fig. S8a) and the other grain is cut from the previous single-crystal one (i.e., 5-nm cube with $[-\frac{1}{2}0\frac{1}{2}]$, [010], $[\frac{1}{2}0\frac{1}{2}]$ directions as shown in Fig. S8b). Following this procedure, samples consisting of two grains without SRO (the 0 sample) and with SRO (the 4M sample) can be constructed. To introduce SRO at the grain boundaries between two 4M samples, 10^5 MC/MD iterations were performed in the shaded regions marked in Fig. S8c (1 nm thickness) to reduce the system potential energy. The models consisting of two grains without SRO (the 0-0 sample) and with SRO (the 4M-4M sample) are shown in Figs. S8d and e. Figure S9a plots the phase transformation fraction and stress variation with strain for these two samples in a 300K tensile test. The 4M-4M sample (continuous red line) has a higher ultimate tensile stress than the 0-0 sample (dashed red line), indicating that SRO still improves the mechanical properties of this HEA with grain boundaries.

The variation of phase transformation with strain plotted in Fig. S9a shows that the critical strain for phase transformation in both the 4M-4M (continuous blue line) and 0-0 samples (dashed blue line) is 1.8%, corresponding to the critical stress of 1.3 GPa (continuous and dashed red lines). These critical strain and stress (1.8% and 1.3 GPa) are lower than those of single-crystal samples (4M sample: 2.6% and 1.6 GPa; 0 sample: 4% and 2.3 GPa, as shown in Fig. S6a). Atomic configurations of the 0-0 and 4M-4M samples at a 1.8% strain are drawn in Figs. S9b and c, respectively, demonstrating that the phase transformation nucleates at the grain boundaries. Hence, the grain boundaries lower the critical stress required for the nucleation of phase transformation. The phase transformation in the 0-0 sample propagates through the grain interior and along the grain boundaries with increasing the strain (as exhibited in Figs. S9b and c). However, the propagation of phase transformation throughout the grain interior and along the grain boundaries in the 4M-4M sample is prohibited by highly-stable FCCP clusters formed in the grain interior and SROs formed in the grain boundaries (as shown in Figs. S9d and e). Besides, the SROs formed in the grain boundaries of 4M-4M sample increase the critical stress required for dislocation nucleation (4M-4M sample: 2.6 GPa, 0-0 sample: 2.2 GPa), leading to the enhancement of the ultimate strength.

In the revised manuscript, we have added the following figures and text to analyze and discuss the grain-boundary effect.

Fig. S8 Construction and atomic configurations of the CoCuFeNiPd HEAs with grain boundaries. **a** Grain 1 with orientations: $[100]$, $[010]$, and $[001]$ and **b** Grain 2 with orientations: $[-\frac{1}{2} 0 \frac{1}{2}]$, $[010]$, and $[\frac{1}{2} 0 \frac{1}{2}]$ along the $[010]$ direction meet to form a sample **c** with grain boundaries. Atomic configurations of the CoCuFeNiPd HEAs consisting of **d** two 0 grains (the 0-0 sample), and **e** two 4M grains after 10^5 iterations for grain boundaries (the 4M-4M sample).

Fig. S9 Fraction of phase transformation, stress vs strain curves and atomic configurations of the CoCuFeNiPd HEAs with grain boundaries during tension at 300 K. **a** Variations of phase transformation fraction and stress with strain for the 0-0 and 4M-4M samples. Atomic configurations of the 0-0 sample at **b** 1.8% strain (0-0-1.8%) and **c** 5.4% strain (0-0-5.4%), and the 4M-4M sample at **d** 1.8% strain (4M-4M-1.8%) and **e** 4.4% strain (4M-4M-4.4%) colored according to their common neighbor analysis phase structures.

Page 17: “We performed additional simulations of bicrystal samples with grain boundaries, as shown in Figs. S8 and S9. The bicrystal sample with SRO also exhibits enhanced ultimate strength and ductility; the associated results are presented and discussed in Supplementary Information.”

Page S4: “To further study the effect of SRO on the mechanical properties of this HEA with grain boundaries, a model of two grains in the same size but with different orientations was constructed by aligning them along the [010] direction (Fig. S8c), where one grain is identical to the previous single-crystal one (i.e., a 5-nm cube with [100], [010], and [001] directions, as shown in Fig. S8a), and the other grain is cut from the previous single-crystal one (i.e., a 5-nm cube with $[-\frac{1}{2} 0 \frac{1}{2}]$, [010], and $[\frac{1}{2} 0 \frac{1}{2}]$ directions, as presented in Fig. S8b). Following this procedure, samples consisting of two grains without SRO (the 0 sample) and with SRO (the 4M sample) can be constructed. To introduce SRO at grain boundaries between two 4M samples, 10^5 MC/MD iterations were performed in the shaded regions marked in Fig. S8c (1 nm thickness) to reduce the system potential energy. The models without SRO and with SRO are shown in Fig. S8d (0-0 sample) and Fig. S8e (4M-4M sample), respectively.

Variations of phase transformation and stress with strain for these two samples during tension at 300 K (along [010] direction) are plotted in Fig. S9a. It is seen that the 4M-4M sample (continuous red line) has a higher ultimate tensile stress than the 0-0 sample (dashed red line), indicating that SRO still improves the mechanical properties of this HEA with grain boundaries. The variation of phase transformation with strain plotted in Fig. S9a shows that the critical strain for phase transformation in both the 4M-4M (continuous blue line) and 0-0 samples (dashed blue line) is 1.8%, corresponding to the critical stress of 1.3 GPa (continuous

and dashed red lines). These critical strain and stress (1.8% and 1.3 GPa) are lower than those of single-crystal samples (4M sample: 2.6% and 1.6 GPa; 0 sample: 4% and 2.3 GPa in Fig. S6a). Atomic configurations of the 0-0 and 4M-4M samples at a 1.8% strain shown in Figs. S9b and c, respectively, demonstrate that the phase transformation nucleates at the grain boundaries. Therefore, the grain boundaries lower the critical stress required for the nucleation of phase transformation. The phase transformation in the 0-0 sample propagates toward the grain interior and along the grain boundaries with increasing the strain (as shown in Figs. S9b and c). However, the propagation of phase transformation toward the grain interior and along the grain boundaries in the 4M-4M sample is prohibited by highly stable FCCP clusters formed in the grain interior and SROs in the grain boundaries (as exhibited in Figs. S9d and e). Besides, the SROs formed in the grain boundaries of 4M-4M sample increase the critical stress required for dislocation nucleation (4M-4M sample: 2.6 GPa, 0-0 sample: 2.2 GPa), leading to the enhancement of the ultimate strength.”

Comment 13: *One of the questions that emerges is perhaps can the authors speculate what would be the morphology of the tensile deformation curves if the strain rate were to be reduced to a more normal level instead of the very high strain rate assumed by the simulations which corresponds to $2 \times 10^8 \text{ s}^{-1}$.*

Author reply: We discussed strain rate effects in response to Comment #3 by this reviewer, above. Unfortunately, other than estimating the change in the tensile strength with strain rate, we are unable to map out the entire stress-strain curve by MD simulations. The computational demand of such a calculation greatly exceeds our resources (or anyone’s we suspect).

Comment 14: *The authors argue that in order to confirm the result that ultimate tensile strength and ductility are simultaneously enhanced via SRO formation is not related to strain rate and hence, they conducted simulations at strain rates that are 10% of the original simulation to support this argument. The question arises as to why was the 10% of the original strain rate used as the baseline to validate the results?*

Author reply: Indeed, we conducted simulations at a strain rate 10% the original to get a sense of the effect of strain rate. Unfortunately, this is the best we can do with available computational resources. As discussed above in response to Comment #3 by this reviewer, we have estimated the change in strength. These two results (10% strain rate and estimating changes in the strength) represent our best attempts at trying to understand the strain rate effects. Unfortunately, MD is not an appropriate choice for low-strain-rate simulations.

Comment 15: *Overall this is a well written paper that provides interesting and novel results of an approach that can be used to design HEAs in the future. Their results suggest that the*

development of SRO can induce structural variations on a sufficiently small scale to achieve outstanding mechanical properties. The authors are all well-known in their respective fields and it is my opinion that the paper as written, with some clarifications, represents an outstanding contribution to the literature.

In view of the above recommend acceptance once the above comments are addressed.

Author reply: We thank the reviewer for her/his strong support and thoughtful insights. We have performed additional simulations/calculations/theoretical analyses and revised our manuscript to reflect both the spirit and substance of her/his comments and suggestions.

REVIEWER COMMENTS

Reviewer #1 (Remarks to the Author):

The authors mostly addressed the issues I raised properly. There are a few additional issues related to their response.

(1) Reliability of atomic potential data

I can understand that it is very difficult to validate the results in the manuscript experimentally at this moment. However, I often see atomistic simulation studies that are inconsistent with experimental facts, even in many high-impact journals like Nature. In my understanding, such discrepancies originate from atomic potential data used for calculations. Is there any method to check the reliability of the atomic potential used in the study? For instance, the authors may be able to check whether the melting point of the HEA is in a reasonable range.

(2) On the location of the phase boundaries

This is just my interest. I guess the boundary between the parts having different thermodynamical stability is diffuse according to the figures in the manuscript. If so, how do the phase boundaries between FCC and BCC move? Do they stop moving at specific locations having specific energy differences between the phases? or something different determines the position of the phase boundaries?

If the authors can answer these (particularly (1)), I can recommend the manuscript for publication.

Reviewer #2 (Remarks to the Author):

Dear Editor,

The authors have addressed all my previous questions by adding quite a lot of new analyses and discussions.

Two more points.

1.

For the 4M+ case, I think it is too close to the 4M case, where the difference is just the fluctuations. There are 8788 atoms in the MC supercell, so 10000 swaps might be too few to get a completely independent structure.

The annealing at 1200K (Fig. S11) suggests that the SRO in CoCuFeNiPd is not always improving the strength and ductility. So in practice, it is very critical to control the heat treatment - to get the "best"

SRO, which then leads to the "best" mechanical properties via the composite-based mechanism.

The authors may revise some statements if they agree with the comments above.

2.

In the calculation of the "phase stability" step 1 (10626 compositions in the .xlsx file), how were the atomic and supercell relaxations performed in DFT?

In the DFT calculations, how were the magnetic moments initialized?

Please indicate these details in the text.

Otherwise, I don't have other questions.

Best wishes,
Binglun Yin

Reviewer #3 (Remarks to the Author):

Thank you for addressing my concerns; I am pleased with the revised version and recommend it for publication.

Response to Reviewer #1

Overall Comments: *The authors mostly addressed the issues I raised properly. There are a few additional issues related to their response.*

Author reply: We thank the reviewer for her/his constructive suggestions/comments on our work. We address these below and indicate where changes were made in the revised manuscript.

Technical Comments:

Comment 1: *Reliability of atomic potential data*

I can understand that it is very difficult to validate the results in the manuscript experimentally at this moment. However, I often see atomistic simulation studies that are inconsistent with experimental facts, even in many high-impact journals like Nature. In my understanding, such discrepancies originate from atomic potential data used for calculations. Is there any method to check the reliability of the atomic potential used in the study? For instance, the authors may be able to check whether the melting point of the HEA is in a reasonable range.

Author reply: We thank the reviewer for her/his understanding of the difficulties of experimental validation at this moment. Following the reviewer's suggestion, we have further checked the reliability of the atomic potential used in the study from three different aspects: 1. lattice constants, 2. cohesive energies and 3. melting points.

The lattice constant of CoCrFeNiPd high-entropy alloy (HEA) was measured experimentally by Ding et al. [Nature 574, 223–227 (2019)] and found to be 3.67 Å. It was reported that the effect of Cu on the lattice constant of CoCrFeNiCu_x (x=0, 0.2, 0.4, 0.6, 0.8, 1.0) HEAs was negligible [Scr. Mater. 161, 28–31 (2019)] due to the similar atomic radius of Cu with Co, Cr, Fe and Ni. Hence, it is expected that the lattice constant of CoCuFeNiPd should be also around 3.67 Å. To make a comparison, we calculated the average lattice constants of the CoCuFeNiPd HEAs after 0 (0 sample), 2×10⁶ (2M sample) and 4×10⁶ (4M sample) iterations and found that they were 3.674 Å, 3.667 Å and 3.666 Å, respectively. Clearly, these predicted values using the atomic potential match the expected value well.

We have also compared the cohesive energies of L1₂ AB₃ alloys for all binary combinations of {Co, Cu, Fe, Ni and Pd} calculated using the density-functional theory (DFT) and the atomic potential (Table S2), and a good agreement is observed. More specifically, both results indicate that the Cu-Cu pairs exhibit the highest cohesive energy, Cu₃-containing pairs (shaded in red) always have higher cohesive energies than other pairs, Co₃/Fe₃/Ni₃-containing pairs (shaded in green) always have lower cohesive energies than other pairs, and Pd₃-containing pairs (shaded in orange) are of medium energies.

Finally, we have performed additional simulations on the heating of the 0, 2M and 4M samples

from 300 K to 2300 K at a heating rate of 100 K/ns. The variations of potential energy with temperature for these three samples are shown in Fig. S14 (A dramatic increase in potential energy corresponds to the starting point of melting), which indicates the melting points are 1831 K, 1750 K and 1685 K, respectively. Experimental measurements reported that the melting points of CoCrFeNi, CoCrCuFe and CoCrCuFeNi are 1711 K, 1623 K and 1662 K [J. Alloys Compd. 783, 193–207 (2019); Intermetallics 86, 59–72 (2017); Metall. Mater. Trans. A 36, 881–893 (2005)], and that of pure Pd is 1828 K [Scr. Mater. 126, 29–32 (2017)]. Therefore, the melting points of the CoCuFeNiPd HEAs in this study are in a reasonable range with respect to the experimental values.

Clearly, the good agreement in the lattice constants, cohesive energies and melting points obtained using the atomic potential and from other sources validates the reliability of the atomic potential used in the study.

In the revised main text, we have added the following sentences:

Page 25: “We have checked the reliability of the atomic potential used in the study from three different aspects: 1. lattice constants, 2. cohesive energies and 3. melting points. A good agreement in the lattice constants, cohesive energies (Table S2) and melting points (Fig. S14) obtained using the atomic potential and from other sources is observed, thus validating the reliability of the atomic potential used in the study (detailed analysis and comparison are presented in Supplementary Information).”

In addition, we have also added the following table, figure and discussions in the Supplementary Information to present the detailed analysis and comparison.

Table S2. Comparison of cohesive energies of the $L1_2$ unit cell. a MD simulations. b DFT calculations.

Co ₃	-4.39	-4.16	-4.35	-4.41	-4.24	Co ₃	-5.16	-4.63	-5.13	-5.09	-4.72
Cu ₃	-3.73	-3.54	-3.69	-3.75	-3.63	Cu ₃	-3.73	-3.49	-3.66	-3.81	-3.63
Fe ₃	-4.25	-4.02	-4.19	-4.26	-4.19	Fe ₃	-4.97	-4.52	-4.88	-4.94	-4.61
Ni ₃	-4.43	-4.19	-4.38	-4.45	-4.25	Ni ₃	-4.89	-4.45	-4.96	-4.82	-4.49
Pd ₃	-4.01	-3.82	-4.05	-4.00	-3.91	Pd ₃	-3.96	-3.66	-4.08	-3.90	-3.63
a	Co	Cu	Fe	Ni	Pd	b	Co	Cu	Fe	Ni	Pd

Fig. S14 Variation of potential energy with temperature for CoCuFeNiPd HEA after 0, 2×10^6 (2M) and 4×10^6 (4M) iterations.

Page S8: “We have checked the reliability of the atomic potential used in the study from three different aspects: 1. lattice constants, 2. cohesive energies and 3. melting points.

The lattice constant of CoCrFeNiPd HEA was measured experimentally by Ding et al.^{S5} and found to be 3.67 Å. It was reported that the effect of Cu on the lattice constant of CoCrFeNiCu_x (x=0, 0.2, 0.4, 0.6, 0.8, 1.0) HEAs was negligible^{S6} due to the similar atomic radius of Cu with Co, Cr, Fe and Ni. Hence, it is expected that the lattice constant of CoCuFeNiPd should be also around 3.67 Å. To make a comparison, we calculated the average lattice constants of the 0, 2M, and 4M samples and found that they were 3.674 Å, 3.667 Å and 3.666 Å, respectively. Clearly, these predicted values using the atomic potential match the expected value well.

We have also compared the cohesive energies of $L1_2 AB_3$ alloys for all binary combinations of {Co, Cu, Fe, Ni and Pd} calculated using the density-functional theory (DFT) and the atomic potential (Table S2), and a good agreement has been observed. More specifically, both results indicate that the Cu-Cu pairs exhibit the highest cohesive energy, Cu₃-containing pairs (shaded in red) always have higher cohesive energies than other pairs, Co₃/Fe₃/Ni₃-containing pairs (shaded in green) always have lower cohesive energies than other pairs, and Pd₃-containing pairs (shaded in orange) are of medium energies.

Finally, we have performed additional simulations on the heating of the 0, 2M, and 4M samples from 300 K to 2300 K in a heating rate of 100 K/ns. The variations of potential energy with temperature for these three samples are shown in Fig. S14 (A dramatic increase in potential energy corresponds to the starting point of melting), which indicates the melting points are 1831 K, 1750 K and 1685 K, respectively. Experimental measurements reported that the melting points of CoCrFeNi, CoCrCuFe and CoCrCuFeNi are 1711 K^{S7}, 1623 K^{S8} and 1662 K^{S9}, and that of pure Pd is 1828 K^{S10}. Therefore, the melting points of the CoCuFeNiPd HEAs in this study are in a reasonable range with respect to the experimental values.

Clearly, the good agreement in the lattice constants, cohesive energies and melting points

obtained using the atomic potential and from other sources validates the reliability of the atomic potential used in the present study.”

Comment 2: *On the location of the phase boundaries*

This is just my interest. I guess the boundary between the parts having different thermodynamical stability is diffuse according to the figures in the manuscript. If so, how do the phase boundaries between FCC and BCC move? Do they stop moving at specific locations having specific energy differences between the phases? or something different determines the position of the phase boundaries?

Author reply: We thank the reviewer for the insightful comment. Following the reviewer’s suggestion, we performed additional analyses on the evolution of a BCC domain (Fig. S13). The variation of the number of BCC atoms in this domain with strain is plotted in Fig. S13a, and the corresponding atomic configurations are drawn in Figs. S13 b to g. These results indicate that the BCC domain expands and phase boundaries move towards FCC phase prior to dislocation nucleation (Figs. S13b to f) due to the increasing phase transformation. After dislocations are nucleated (due to the elastic interactions of FCC and BCC), the BCC domain shrinks and phase boundaries move towards BCC phase due to the stress reduction and dislocation nucleation/propagation (Figs. S13f to g).

In the revised manuscript, we have added the following figure and discussions to emphasize these points.

Fig. S13 Evolution of a BCC domain in the CoCuFeNiPd HEA with 2×10^6 iterations at 300 K (the 2M sample) during tension at 300 K. a Variation of the number of BCC atoms in the domain with strain. **b-g** Atomic configurations of the BCC domain at different strains (marked in **a**) according to their common neighbor analysis on phase structures.

Page 24: “The evolution of a BCC domain is shown in Fig. S13. It is seen that the BCC domain expands and phase boundaries move towards FCC phase prior to dislocation nucleation (Figs. S13b to f) due to the increasing phase transformation. After dislocations are nucleated (due to

the elastic interactions of FCC and BCC), the BCC domain shrinks and phase boundaries move towards BCC phase (Figs. S13f to g) due to the stress reduction and dislocation nucleation/propagation.”

Comment 3: If the authors can answer these (particularly (1)), I can recommend the manuscript for publication.

Author reply: We would like to thank the reviewer again for her/his valuable comments and suggestions. We performed additional simulations/analyses and revised our manuscript to reflect both the spirit and substance of the reviewer’s suggestions/questions; doing so has validated the reliability of the atomic potential and improved the clarity of our manuscript. We trust that these changes will be satisfactory to the reviewer.

Response to Reviewer #2

Overall Comments: The authors have addressed all my previous questions by adding quite a lot of new analyses and discussions.

Author reply: We thank the reviewer for his recognition of our revisions and constructive suggestions/comments on our work. We respond each of the suggestions/comments from the reviewer below, as well as indicate how we revised the manuscript.

Technical Comments: Two more points.

Comment 1: For the 4M+ case, I think it is too close to the 4M case, where the difference is just the fluctuations. There are 8788 atoms in the MC supercell, so 10000 swaps might be too few to get a completely independent structure.

The annealing at 1200K (Fig. S11) suggests that the SRO in CoCuFeNiPd is not always improving the strength and ductility. So in practice, it is very critical to control the heat treatment - to get the "best" SRO, which then leads to the "best" mechanical properties via the composite-based mechanism.

The authors may revise some statements if they agree with the comments above.

Author reply: We agree with the reviewer that the 4M+ case might not be a completely independent structure from the 4M case. This can be understood from the fact that the 4M+ is evolved from the 4M case. But they do show apparent differences, for example, the fractions of FCC-preferred (FCCP) clusters and BCC-preferred (BCCP) clusters are 33% and 9% in the 4M+ case, 32% and 8% in the 4M case, and 31% and 7% in the 2M case. Therefore, the differences in the FCCP and BCCP fractions between the 4M+ and 4M cases are as much as those between the 4M and 2M cases. Thus, we believe the differences are beyond just the fluctuations. In this calculation, 10000 swaps are sufficient to obtain distinct differences since each swap is only accepted when the sum of squares of the Warren-Cowley parameters is increased. In our previous calculations (from 0 to 4M case), the acceptance of each swap conformed to the Metropolis criterion, i.e., the swap with probability higher than / equal to randomly-generated number was also accepted. We also followed the Metropolis criterion to perform swaps from the 4M case but could not obtain 33% FCCP and 9% BCCP (as those in 4M+ case) after 2 more million iterations.

We agree with the reviewer that the SRO in CoCuFeNiPd may not be always improving the strength and ductility as the annealing at 1200K (Fig. S11) indicates. Therefore, it is critical to control the heat treatment in order to get the "best" SRO, which then leads to the "best" mechanical properties via the composite-based mechanism.

In the revised manuscript, we have added the following text to emphasize these points.

Page S3: “The “FCCP” and “BCCP” fractions are 33% and 9% in the 4M+ sample, 32% and 8% in the 4M sample, and 31% and 7% in the 2M sample. Therefore, the differences in the “FCCP” and “BCCP” fractions between the 4M+ and 4M samples are as much as those between the 4M and 2M samples. In our previous calculations (from 0 to 4M samples), the acceptance of each swap conformed to the Metropolis criterion, i.e., the swap with probability higher than / equal to randomly-generated number was also accepted. However, from 4M to 4M+ samples, each swap was only accepted when the sum of squares of the WCPs was increased, leading to the distinct difference after 10^4 successful MC iterations.”

Page S7: “Therefore, it is critical to control the heat treatment to get the "best" SRO, which then leads to the "best" mechanical properties via the composite-based mechanism.”

Comment 2: *In the calculation of the "phase stability" step 1 (10626 compositions in the .xlsx file), how were the atomic and supercell relaxations performed in DFT?*

In the DFT calculations, how were the magnetic moments initialized?

Please indicate these details in the text.

Author reply: We thank the reviewer for the useful comment. The calculations of the "phase stability" (i.e., 10626 compositions in the .xlsx file) were done by MD instead of DFT due to the large cell size and expensive computational cost. The initial size of the supercell was $20 \times 20 \times 20 \text{ nm}^3$, where the atoms were relaxed using a conjugate gradient minimization of the potential energy (zero stress in all three $\langle 100 \rangle$ directions).

The cohesive energies of all A-B (and A-A) elemental pairs for binary combinations of {Co, Cu, Fe, Ni and Pd} were calculated using DFT (Table S1). Relaxation runs were done by performing spin-polarized calculations (setting ISPIN = 2 in VASP) and by allowing both the ionic positions, cell volume and cell shape to relax (setting ISIF = 3 in VASP). The initial magnetic moment per atom is 5.0 for Co, Fe and Ni and 0.6 for Cu and Pd, which are consistent with the values taken by Materials Projects (<https://materialsproject.org/>).

In the revised manuscript, we have added the following text to indicate these details.

Page 26: “where the atoms were relaxed using a conjugate gradient minimization of the potential energy (zero stress in all three $\langle 100 \rangle$ directions).”

Page 28: “Relaxation runs were done by performing spin-polarized calculations and by allowing both the ionic positions, cell volume and cell shape to relax.”

Comment 3: *Otherwise, I don't have other questions.*

Author reply: We would like to thank the reviewer again for his valuable comments and suggestions, which certainly help enhance the quality of our work.

Response to Reviewer #3

Overall Comments: Thank you for addressing my concerns; I am pleased with the revised version and recommend it for publication.

Author reply: We thank the reviewer for her/his insightful suggestions/comments and recommendation of our manuscript for publication.

REVIEWERS' COMMENTS

Reviewer #1 (Remarks to the Author):

The authors have addressed the issues I raised properly. Therefore, I can recommend this manuscript for publication in Nature Communications.

Best regards,
Shuhei Yoshida, Ph.D.
Assistant Professor, Kyoto University, Japan.

Reviewer #2 (Remarks to the Author):

Dear Editor,

I am pleased with the current version and do not have other questions.

Best wishes,
Binglun Yin